# Granger Components Analysis: Unsupervised learning of latent temporal dependencies

**Jacek P. Dmochowski**
Department of Biomedical Engineering
City College of New York
New York, NY 10031
jdmochowski@ccny.cuny.edu

## Abstract

A new technique for unsupervised learning of time series data based on the notion of Granger causality is presented. The technique learns pairs of projections of a multivariate data set such that the resulting components – "driving" and "driven" – maximize the strength of the Granger causality between the latent time series (how strongly the past of the driving signal predicts the present of the driven signal). A coordinate descent algorithm that learns pairs of coefficient vectors in an alternating fashion is developed and shown to blindly identify the underlying sources (up to scale) on simulated vector autoregressive (VAR) data. The technique is tested on scalp electroencephalography (EEG) data from a motor imagery experiment where the resulting components lateralize with the side of the cued hand, and also on functional magnetic resonance imaging (fMRI) data, where the recovered components express previously reported resting-state networks.

## 1   Introduction

Unsupervised component analysis techniques such as Independent Components Analysis (ICA) and Canonical Correlation Analysis (CCA) are routinely employed for dimensionality reduction, blind source separation, and feature selection [1, 2]. In particular, these methods are often applied to data from complex systems such as the brain and financial markets, as their interpretability facilitates insight into the structure and dynamics of the system. Nevertheless, the sources underlying complex systems often violate the assumptions made by these classical tools. For example, the sources of activity in the brain are highly dependent due to functional connectivity, and their dynamics exhibit lagged (in addition to instantaneous) temporal correlations.

Granger causality [3] is a well-known technique for measuring a form of dependence rooted in the temporal precedence effect. Time series $x(t)$ is said to "Granger-cause" $y(t)$ if the past of $x$ improves the prediction of the present value of $y$ above that of its own past. The technique has many generalizations [4, 5, 6, 7, 8, 9, 10, 11] and is conventionally utilized as a statistical test. One specifies the two time series being probed and their assumed directionality, and then carries out hypothesis testing on the measured relationship. The time series are typically defined in the sensor space, meaning that if the sources of interest are mixed in the observations, Granger causality may not optimally identify the true relationship. Relatedly, statistically independent sources that are linearly mixed at the sensors may spuriously indicate Granger causality [12, 13, 14].

Here the concept of Granger causality is employed to propose a new criterion for unsupervised learning that is appropriate in the case of temporally-dependent source signals. The basic idea is to identify two projections of a multivariate time series such that the Granger causality among the resulting pair of components is maximized. A coordinate descent algorithm that decomposes a multivariate dataset into pairs of components is proposed such that each pair consists of a "driving"

37th Conference on Neural Information Processing Systems (NeurIPS 2023).

(Granger-causing) and a "driven" (Granger-caused) signal. To demonstrate proof-of-concept, the proposed method is tested on simulated data following the vector autoregressive (VAR) model that is frequently employed in studies of Granger causality [15]. Experiments on recordings from the human brain – previously collected scalp electroencephalogram (EEG) and functional magnetic resonance imaging (fMRI) datasets – are then performed. The technique is evaluated by analyzing and interpreting the recovered components, in particular their relation to what is known about the underlying systems from the human neuroscience literature.

The contributions of this work are:

- A new approach to blind source separation that is appropriate when the desired signals exhibit temporal dependencies.
- A grouped coordinate descent algorithm that learns pairs of signals with a Granger causal relationship and is shown to identify the underlying sources (up to scale) on simulated data.
- Experimental findings from the human brain demonstrating that the technique recovers components whose structure is consistent with earlier findings from neuroscience.

## 2 Related work

The classical approach to inferring latent temporal relationships is CCA [16] and its extensions, for example kernel [17] and Deep CCA [18]. Both CCA and the proposed approach yield pairs of latent variables exhibiting temporal dependencies. In both cases, there is a natural ordering of the component pairs (i.e., magnitude of correlation in CCA, strength of Granger causality in the present work). The key differences between CCA and the proposed method are that: (i) whereas CCA operates on two pre-defined "views" of the data, the proposed method learns latent relationships underlying a single multivariate record, and (ii) the proposed method extracts signals related by Granger causality as opposed to instantaneous correlation (Fig 1).

Multivariate Granger causality [9] is an approach that extends the measure of Granger causality to two sets of variables. As with CCA, the two sets (e.g. regions of interest in the human brain) are first defined, followed by a statistical procedure that assays the relationship between the multivariate records. Several previous works have combined Granger causality with CCA [19, 20, 21]. These techniques employ the CCA framework to eliminate the need for explicitly computing autoregressive models. In all cases, the two sets of variables (or views of the data) are first defined, followed by the assessment of Granger causality. Approaches that combine VAR modeling with PCA [22] and ICA [23] have also been proposed.

## 3 Problem formulation

The following linear signal model is assumed:
$$\mathbf{x}(t) = \mathbf{A}\mathbf{s}(t) + \mathbf{e}(t), \tag{1}$$
where $\mathbf{x}(t) \in \mathbb{R}^D$ is an observed, wide-sense stationary multivariate time series, $\mathbf{A} \in \mathbb{R}^{D \times K}$ is a mixing matrix relating the latent sources $\mathbf{s}(t) = [\; s_1(t) \quad \ldots \quad s_K(t) \;] \in \mathbb{R}^K$ to the observations, and $\mathbf{e}(t) \in \mathbb{R}^D$ is an additive error term representing sensor noise as well as deviations from the assumed signal model.

The latent sources are assumed to exhibit a temporal dependence according to:
$$s_{\mathcal{I}_i}(t) \to s_{\mathcal{J}_i}(t), \quad i \in 1, 2, \ldots, P, \tag{2}$$
where $\to$ denotes "Granger-causality", meaning that the past of $s_{\mathcal{I}_i}(t)$ is linearly predictive of the present of $s_{\mathcal{J}_i}(t)$ beyond that which can be explained by the past of $s_{\mathcal{J}_i}(t)$, and $P$ is the number of temporally dependent latent signal pairs. The mappings $\mathcal{I}_i$ and $\mathcal{J}_i$ yield the indices of the driving and driven signals, respectively, of the $i$th pair of latent sources.

The goal is to recover pairs of signals $y_i(t) = \mathbf{w}_i^T \mathbf{x}(t)$ and $z_i(t) = \mathbf{v}_i^T \mathbf{x}(t)$ such that:
$$\begin{aligned} y_i(t) &\approx s_{\mathcal{I}_i}(t), \\ z_i(t) &\approx s_{\mathcal{J}_i}(t), \quad i = 1, 2, \ldots P. \end{aligned}$$

Below, an objective based on the Granger causality between pairs of component signals is proposed.

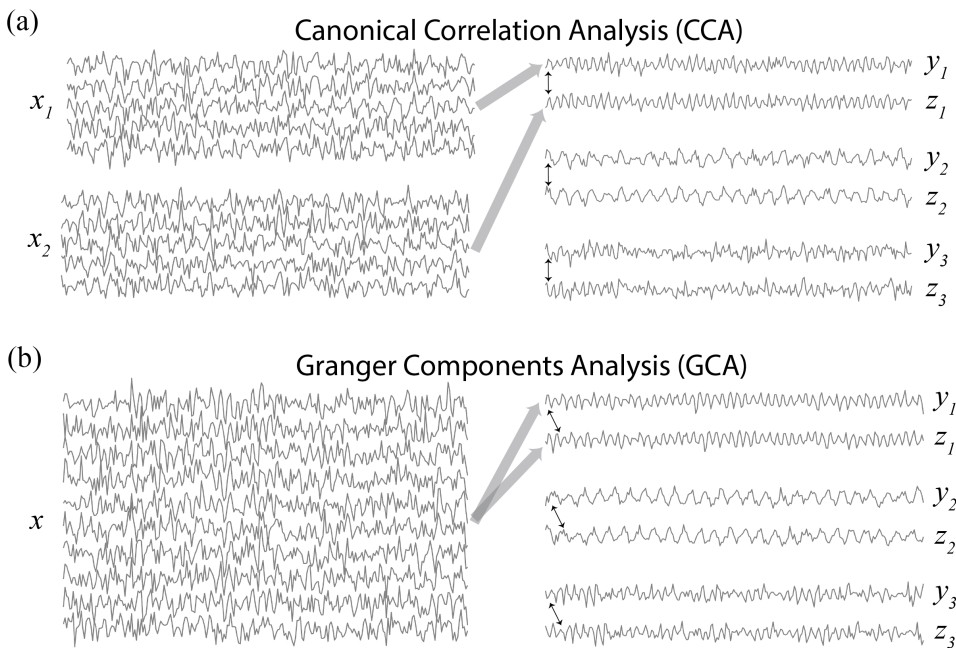

Figure 1: Relating the proposed method to Canonical Correlation Analysis (CCA). **(a)** CCA learns correlated subspaces from two multivariate data sets. **(b)** Granger Components Analysis (GCA) operates on a single data set and yields pairs of components exhibiting Granger causal relationships.

## 4   Granger Components Analysis

Throughout, a history length of $L$ samples is assumed. Denote $\epsilon_z$ as the residual of a linear regression from $\mathbf{z}_p(t) = [z(t-1), \ldots, z(t-L)]$ onto $z(t)$, and $\epsilon_{zy}$ as the residual of a regression from $[\mathbf{z}_p(t), \mathbf{y}_p(t)]$ onto $z(t)$, where $\mathbf{y}_p(t) = [y(t-1), \ldots, y(t-L)]$. The goal is to identify latent component time series $y(t)$ and $z(t)$ such that the Granger causality from $y$ to $z$ is maximized:

$$\mathbf{w}^*, \mathbf{v}^* = \arg \min_{\mathbf{w}, \mathbf{v}} \log \frac{\langle \epsilon_{zy}^2 \rangle}{\langle \epsilon_z^2 \rangle}. \tag{3}$$

The objective function (3) is closely related to the logarithm of the F-statistic between the "full" and "reduced" regression models inherent to Granger causality [15]. The smaller the ratio of residual powers, the more information is contained in the past of driving signal $y$ about the present of driven signal $z$.

To relate the objective function to the coefficient vectors $\mathbf{w}$ and $\mathbf{v}$, the mean residuals are written as (see *Supplementary Material*):

$$\langle \epsilon_{zy}^2 \rangle = \sigma_z^2 - \mathbf{r}^T \mathbf{R}^{-1} \mathbf{r},$$
$$\langle \epsilon_z^2 \rangle = \sigma_z^2 - \mathbf{q}^T \mathbf{Q}^{-1} \mathbf{q},$$

where the following covariance vectors and covariance matrices have been defined:

$$\mathbf{r} = \left( \mathbf{I}_{2L} \otimes \mathbf{v}^T \right) \left( \mathbf{I}_2 \otimes \mathbf{\Sigma}_{1:L} \right) \begin{pmatrix} \mathbf{1}_L \otimes \mathbf{v} \\ \mathbf{1}_L \otimes \mathbf{w} \end{pmatrix} \qquad \mathbf{q} = \left( \mathbf{I}_L \otimes \mathbf{v}^T \right) \mathbf{\Sigma}_{1:L} \left( \mathbf{1}_L \otimes \mathbf{v} \right)$$

$$\mathbf{R} = \begin{pmatrix} \mathbf{1}_2^T \otimes \mathbf{I}_L \otimes \mathbf{v}^T \\ \mathbf{1}_2^T \otimes \mathbf{I}_L \otimes \mathbf{w}^T \end{pmatrix} \left( \mathbf{I}_2 \otimes \tilde{\mathbf{\Sigma}} \right) \begin{pmatrix} \mathbf{I}_L \otimes \mathbf{v} & \mathbf{0} \\ \mathbf{0} & \mathbf{I}_L \otimes \mathbf{w} \end{pmatrix} \qquad \mathbf{Q} = \left( \mathbf{I}_L \otimes \mathbf{v} \right)^T \tilde{\mathbf{\Sigma}} \left( \mathbf{I}_L \otimes \mathbf{v} \right),$$

and where $\otimes$ denotes the Kronecker product, $\mathbf{1}_M$ is a column vector of $M$ ones, and $\mathbf{I}_M$ is the $M$-by-$M$ identity matrix. The $LD$-by-$LD$ block covariance matrices are given by:

$$\mathbf{\Sigma}_{1:L} = \begin{pmatrix} \mathbf{\Sigma}(1) & \mathbf{0} & \ldots & \mathbf{0} \\ \mathbf{0} & \mathbf{\Sigma}(2) & \ldots & \mathbf{0} \\ \vdots & \mathbf{0} & \ddots & \vdots \\ \mathbf{0} & \ldots & \ldots & \mathbf{\Sigma}(L) \end{pmatrix} \qquad \tilde{\mathbf{\Sigma}} = \begin{pmatrix} \mathbf{\Sigma}(0) & \mathbf{\Sigma}(-1) & \ldots & \mathbf{\Sigma}(-L+1) \\ \mathbf{\Sigma}(1) & \mathbf{\Sigma}(0) & \ldots & \mathbf{\Sigma}(-L+2) \\ \vdots & \mathbf{\Sigma}(1) & \ddots & \vdots \\ \mathbf{\Sigma}(L-1) & \ldots & \ldots & \mathbf{\Sigma}(0) \end{pmatrix},$$

where $\boldsymbol{\Sigma}(\tau) = E\left\{\mathbf{x}(t)\mathbf{x}^T(t-\tau)\right\}$ is the lagged covariance of the observations.

The optimization problem (3) is non-convex because the objective function is invariant to scalings of $\mathbf{w}$ and $\mathbf{v}$. Consequently, multiple pairs of coefficient vectors locally minimize the objective, and it is only possible to recover the latent sources up to a scaling factor.

A known property of Granger causality [9, 10] is that if $y \to z$, then $ay + bz \to cz$, where $a$, $b$, and $c$ are arbitrary constants. This means that, without appropriate modifications to the objective, it is only possible to identify the driven signal $z$. To resolve this ambiguity, the concept of time-reversed Granger causality [24, 12] is utilized. Note that if $y \to z$ in forward time, then $z \to y$ in reversed time. Forward and reversed time are then combined into a single objective function according to:

$$J(\mathbf{w}, \mathbf{v}) = \log \frac{\left\langle \epsilon_{zy}^2 \right\rangle \left\langle \tilde{\epsilon}_{yz}^2 \right\rangle}{\left\langle \epsilon_z^2 \right\rangle \left\langle \tilde{\epsilon}_y^2 \right\rangle}, \tag{4}$$

where $\tilde{\epsilon}$ indicates that the mean residual is computed with time reversed.

## 5  Grouped coordinate descent algorithm

To minimize (4) with respect to the coefficient vectors $\mathbf{w}$ and $\mathbf{v}$, a grouped coordinate descent algorithm that optimizes for $\mathbf{v}$ while holding $\mathbf{w}$ fixed and vice versa is proposed. This approach is commonly employed in problems where the independent variables partition naturally [25].

At each iteration of the algorithm, non-linear constraints of the form $\|\mathbf{v}\| = 1$ (or $\|\mathbf{w}\| = 1$) are added to yield unit-norm solutions. The resulting constrained optimization problem may be solved with conventional interior-point or sequential quadratic programming solvers. During development, MATLAB's "fmincon" and Python's "scipy.optimize" were both tested. An expression for the gradient of the objective function (4) with respect to $\mathbf{v}$ and $\mathbf{w}$ is derived in the *Supplementary Material*. However, it was found that the use of numerical differentiation reduced computational time and was thus employed for all empirical analyses.

As the cost function is non-convex, there are potentially several pairs of components that yield meaningful latent sources. In order to recover $P$ pairs of components $\{y_i(t), z_i(t)\}, i = 1, \ldots, P$, here it is proposed to repeat the optimization after the first iterate, but not before removing the contribution of the driving signal $y_1(t)$ from the data. This takes the form of a spatiotemporal regression such that any signals that are correlated with $y_1(t)$ or its lagged versions $y_1(t-l), l = 1, \ldots, L$ are removed. This procedure is repeated until the desired number of component pairs $P$ is obtained.

The proposed algorithm is described in Algorithm 1, where $\mathbf{X}$ is a $D$-by-$T$ matrix storing the observed data, $\mathbf{Y}_p$ is an $L$-by-$T$ convolution matrix allowing the regression of $\mathbf{y}_p(t)$ onto $\mathbf{X}$, convmatrix is an operation that produces a convolution matrix, $^\#$ denotes the Moore-Penrose pseudoinverse, and $\sigma$ is a small positive constant that is used to randomly initialize the weights of the coefficient vectors.

---

**Algorithm 1** Grouped coordinate descent algorithm for GCA.

---

$\quad p \leftarrow 1$
$\quad$**while** $p \leq P$ **do**
$\quad\quad\quad \mathbf{w}_p \leftarrow \mathcal{N}(\mathbf{0}, \sigma^2\mathbf{I})$
$\quad\quad\quad \mathbf{v}_p \leftarrow \mathcal{N}(\mathbf{0}, \sigma^2\mathbf{I})$
$\quad\quad\quad$**repeat**
$\quad\quad\quad\quad\quad \mathbf{v}_p \leftarrow \arg\min_{\mathbf{v}} J(\mathbf{w}_p, \mathbf{v}, \mathbf{X}) \quad$ subject to $\|\mathbf{v}\| = 1$
$\quad\quad\quad\quad\quad \mathbf{w}_p \leftarrow \arg\min_{\mathbf{w}} J(\mathbf{w}, \mathbf{v}_p, \mathbf{X}) \quad$ subject to $\|\mathbf{w}\| = 1$
$\quad\quad\quad$**until** converged
$\quad\quad\quad \mathbf{y}_p \leftarrow \mathbf{w}_p^T\mathbf{X}$
$\quad\quad\quad \mathbf{Y}_p \leftarrow \text{convmatrix}(\mathbf{y}_p)$
$\quad\quad\quad \mathbf{X} \leftarrow \mathbf{X}\left(\mathbf{I} - \mathbf{Y}_p^\#\mathbf{Y}_p\right)$
$\quad\quad\quad p \leftarrow p + 1$
$\quad$**end while**

---

## 6 Experiments

**Illustrative example: Gaussian VAR system**

As a first step, proof-of-concept simulations were performed to evaluate the method's ability to recover the latent structure embedded in mixed time series. A $D = 10$-dimensional time series with $K = 5$ latent sources whose activity was linearly mixed in the observations (Fig 2a) was generated. The innovation process was distributed according to $\mathcal{N}(\mathbf{0}, \mathbf{I})$. White Gaussian noise with $\sigma = 0.1$ was also added to the sensors. The elements of the mixing matrix were drawn from the uniform distribution over $[0, 1]$. Importantly, three of the sources exhibited temporal dependencies, with $s_1 \rightarrow s_2$ and $s_2 \rightarrow s_3$, while $s_4$ and $s_5$ were independent from all other sources. For convenience, $s_1$-$s_3$ are referred to as "connected" and $s_4$-$s_5$ as "disconnected" sources. The dynamics of the connected sources followed a VAR(3) system with coefficients taken from Stokes and Purdon [26], a study that demonstrated the shortcomings of conventional Granger causality. The disconnected sources also followed a VAR(3) model but with the off-diagonal elements set to 0 (no cross-correlation with other sources). 100 runs of 5000 samples were simulated, and $P = 3$ pairs of Granger components were estimated.

The value of the objective function after convergence $J^*$ is depicted in the vertical axis of Fig 2b (error bars indicate standard errors of the mean across 100 replicates). The true value of the objective (obtained by measuring the Granger causality between $s_1$ and $s_2$ and between $s_2$ and $s_3$) is indicated in red and blue markers for $s_1 \rightarrow s_2$ and $s_2 \rightarrow s_3$, respectively. It is apparent that the method recovered latent dependencies whose magnitude matched the true strength of causality. As expected, the Granger causality between $y_3$ and $z_3$ was near zero. It was then evaluated whether the time series recovered in the first two Granger pairs $(y_1, z_1)$ and $(y_2, z_2)$ corresponded to those of the latent dependent sources $(s_1, s_2)$ and $(s_2, s_3)$, respectively. Indeed, GCA recovered the dynamics of the connected sources with high reliability (mean $r^2 > 0.96$ across 100 replicates, Fig 2d-g). Note that both the $s_1 \rightarrow s_2$ and $s_2 \rightarrow s_3$ relationships were identified, with $s_2$ captured by *both* $y_2$ and $z_1$.

As a comparison to classic techniques, PCA and ICA were also evaluated on this simulated dataset. In both cases, a significantly lower degree of fidelity (approximately 50% and 30% variance explained for PCA and ICA, respectively) was observed in the recovered signals with respect to ground-truth (Fig S1 in *Supplementary Material*).

**Control experiment: independent sources**

As a null control, simulations with statistically independent sources ($s_1 \perp\!\!\!\perp s_2 \perp\!\!\!\perp s_3$) were also conducted. The objective function values of the first three Granger pairs are shown in Fig 2c, where it is apparent that the algorithm did not recover meaningful temporal dependencies. As depicted in Fig 2h-k, the dynamics of the independent latent sources could not be recovered.

**EEG during motor imagery**

Next, the evaluation of GCA on real-world data was undertaken, concentrating on systems with a partially understood underlying structure, ensuring that the results could be interpreted meaningfully. The technique was first applied to publicly available EEG data recorded from $n = 52$ human participants performing a motor imagery task [27]. Subjects were asked to imagine moving either their left or right hand, and three seconds of brain activity following the cue were recorded. To recover components representative of the cohort, a "group GCA" was performed where the covariance matrices $\mathbf{\Sigma}_{1:L}$ and $\tilde{\mathbf{\Sigma}}$ were aggregated across subjects prior to minimizing (4). Tikhonov regularization that reduced the condition number of the block covariance matrices was applied (see *Supplementary Material* for details), and the temporal aperture was set to $L = 16$ samples (0.5 seconds). It was asked whether GCA would recover signals exhibiting known properties of the motor imagery circuitry: (i) contralateralization of brain activity with respect to the cued hand, and (ii) the flow of information from pre-motor (planning) to motor (execution) regions [28].

$P = 2$ pairs of driving and driven signals were computed for each experimental condition: "Left motor imagery" and "Right motor imagery" (Fig 3). To identify the spatial distribution of the driving and driven signals, the "forward models" corresponding to each component were computed [29]. In panels a-d, the spatial topographies of the driving and driven signals are shown on the top left and top right, respectively. The time courses of driving and driven signals are depicted below the scalp maps.

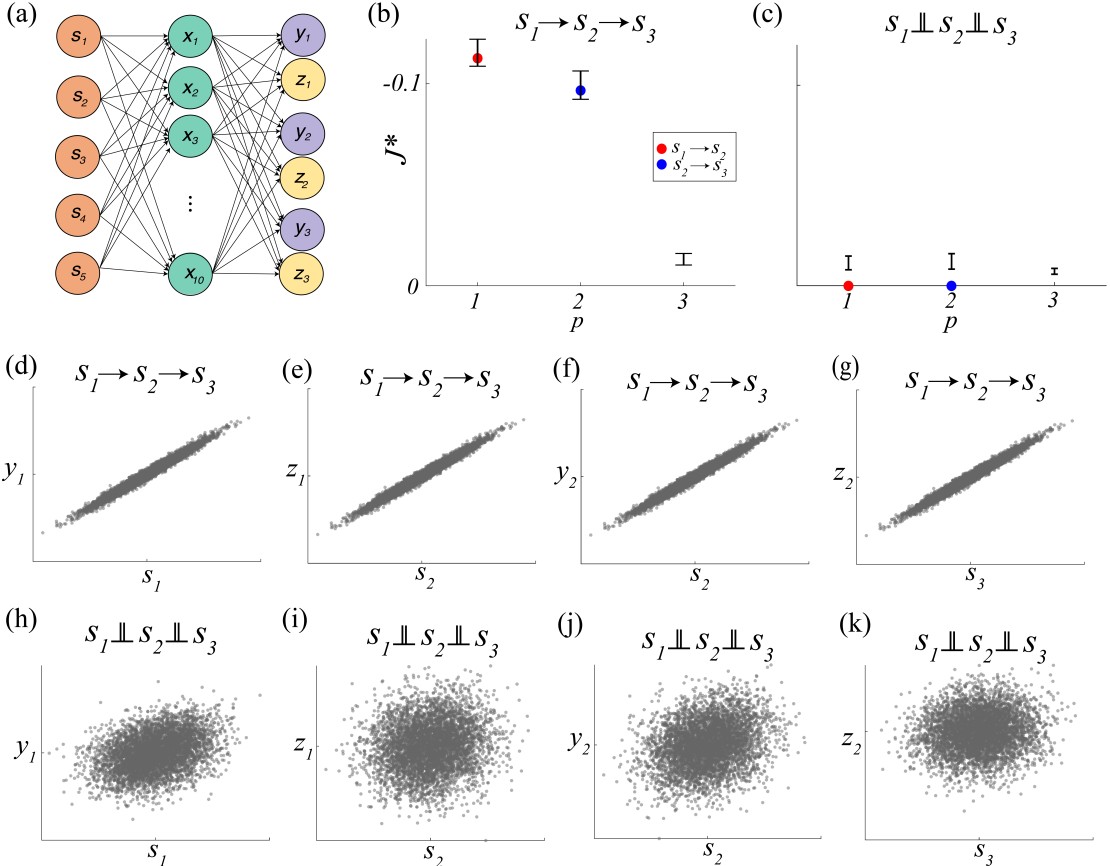

Figure 2: Proof-of-concept simulations with VAR(3) data. **(a)** A $D = 10$-dimensional system with $K = 5$ latent sources that were linearly mixed in the observations along with additive noise was simulated. Two temporal dependencies were modeled: $s_1 \to s_2$ and $s_2 \to s_3$. $P = 3$ pairs of Granger components were subsequently computed. **(b)** The value of the objective function $J^*$ at convergence in each iteration is shown on the vertical axis (inverted for clarity; error bars denote the mean $\pm$ sem across 100 replicates). The true values of the Granger causality are indicated in red and blue markers for $s_1 \to s_2$ and $s_2 \to s_3$, respectively. **(c)** Same as (b) but now shown for control experiments where all sources were independent. **(d)-(g)** Scatter plots depicting the values of the true source signal (horizontal axes) and the recovered component (vertical axes), where each marker is a time sample. In the experiments with temporally dependent source signals, GCA recovered $s_1$ in $y_1$, $s_2$ in both $y_2$ and $z_2$ and $s_3$ in $z_3$ (all $r^2 > 0.96$). **(h)-(k)** In the absence of temporal dependencies, the source signals could not be identified.

For both GC1 and GC2, the components recovered by GCA for left motor imagery were reflected around the midline relative to right motor imagery. The *driven* components of GC1 and GC2 exhibited asymmetric spatial topographies with activation over frontocentral and central electrodes (Fig 3a-d, top right). This pattern is consistent with activation of the motor cortex in one hemisphere. On the other hand, the spatial topographies of the *driving* components were more symmetric, with activation over prefrontal and frontal electrodes (Fig 3a-d, top left), particularly for GC1. This is consistent with the location of anticipatory regions such as the prefrontal cortex and anterior cingulate [30, 31], as well as premotor regions such as the supplementary motor area (SMA) [32].

Interestingly, the components recovered by GC1 were ipsilateral to the cued hand, while those recovered in GC2 exhibited the classical contralateralization of the motor cortex. This ordering may reflect the inhibition of the uncued hand that is required by the task, as well as the *desynchronization* of EEG that is observed during motor activation [33]. For example, the time courses of the components

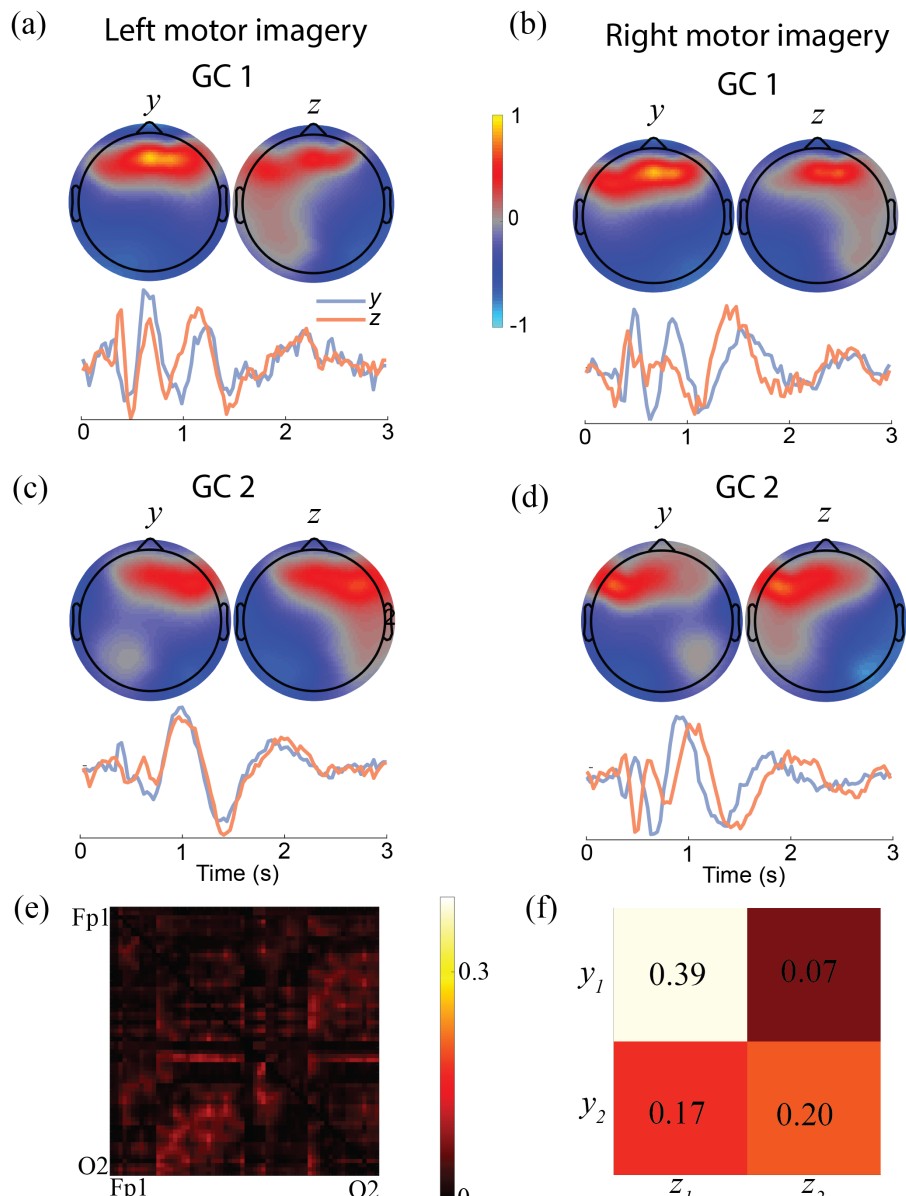

Figure 3: GCA recovers physiologically plausible components from EEG. **(a)** Spatial topographies of the driving (top left) and driven (top right) components of the first Granger pair, as measured during trials in which subjects were asked to imagine moving their left hand. The corresponding time courses are shown below. The driving component is largely symmetric and is expressed over prefrontal and frontal electrodes, consistent with the anticipation and planning of movement. The driven component is asymmetric and shows increased activation over central electrodes on the ipsilateral hemisphere. **(b)** Same as (a) but now for trials where the right hand was cued. The spatial topographies are reflected around the midline compared to left motor imagery, while the time courses are qualitatively similar. **(c)** Topographies and time courses of the second Granger component pair during left imagery, where the activation of the right motor region in the driven signal is consistent with the contralateralization of the activated motor cortex. **(d)** Same as (c) but now for right imagery, where the driven signal shows activation over the left primary motor cortex region. **(e)** Matrix of Granger causality values $(-J)$ between all pairs of electrodes (shown for right motor imagery). The values are limited to $0.12$ and it is difficult to ascertain the Granger causal structure of the system. **(f)** Same as (e) but now for the Granger components. The bottom row hints at a relationship of the form $z_1 \leftarrow y_2 \rightarrow z_2$.

in GC1 exhibited a faster initial oscillation than those in GC2 (Fig 3a-d, bottom panels). From visual inspection of the time courses, the driving signal in each component pair appears to lead the driven signal. Note that while the spatial topographies are flipped with respect to the cued hand, the time courses are qualitatively similar between left and right motor imagery.

The power spectra of the recovered components, as well as those of several individual electrodes, are depicted in Figure S2. Of note, it is interesting that the *driving* signal of GC2 has relatively large 10 Hz "mu" power [33], whereas the corresponding driven signal has relatively low 10 Hz power. This appears to be consistent with the driving signal being preparatory (10 Hz activity is synchronized) and the driven signal reflecting execution (10 Hz desynchronization).

The magnitude of Granger causality between all pairs of electrodes (represented here as -$J$ so that positive values indicate a strong relationship) is depicted in Fig 3e for the right motor imagery condition (see Fig S3 for the results from left motor imagery). The single strongest relationship is 0.12, and it is difficult to interpret the overall pattern of Granger causalities. The matrix of Granger causality strengths for the recovered components is shown in Fig 3f. In addition to larger magnitudes compared to those observed between individual electrodes (as high as 0.39 for $y_1 \rightarrow z_1$), the bottom row of the matrix suggests that $y_2$ drives both $z_1$ and $z_2$. Given the left frontal topography of $y_2$ and the more central topographies of $z_1$ (ipsilateral to cued hand) and $z_2$ (contralateral to cued hand), this is suggestive of an EEG source that projects to both the activated and inhibited sources.

For comparison to existing approaches, the spatial topographies of the components recovered by PCA, ICA, and MVARICA [23] are depicted in Fig S4. The components recovered by the proposed method were not evident in those yielded by these conventional approaches (compare Fig 3 with Fig S4). Moreover, it is notable that the lateralization of the forward models with respect to the side of the cued hand is not readily apparent in Fig S4.

### Resting-state fMRI

At rest, the human brain exhibits stereotyped patterns of activity in the form of resting-state networks (RSN) [34, 35, 36]. Perhaps the most well-known of these is the default-mode network (DMN) [37], a collection of brain regions that appears to become active during introspection and mind-wandering. A previously collected fMRI dataset [38] from $n = 20$ healthy adults in the resting state (eyes open) was employed here to investigate whether GCA could recover components (i.e., sets of brain regions) that reflect the presence of these previously reported networks. As the original study investigated the effects of near-infrared brain stimulation on fMRI activity, only the first 10 minutes of the 30 minute recording (i.e., the segment occurring before the onset of stimulation) was analyzed here. The input data was parcellated into $D = 150$ regions-of-interest (ROIs) following a standard brain atlas [39]. The time series of each ROI was formed by computing the mean of all grey matter voxels in the ROI at each time sample. The temporal aperture was set to $L = 4$ samples ($\approx 11$ s) and the number of components to $P = 6$. Block covariance matrices were pooled across participants prior to analysis, leading to a group analysis. The recovered components were then analyzed by examining the individual brain regions most strongly expressed by the forward model [29] of each driving and driven component.

The five regions with strongest expression in the forward model of each driving and driven component are listed in Table 1. Interestingly, the ROI most strongly expressed in the driving signal of GC1 was the right angular gyrus, one of the three hubs of the DMN [37] (Fig 4a, top row). The left angular gyrus was the fourth strongest driving signal in GC1. The *driven* regions of GC1 were bilateral visual areas and included the primary visual cortex (Fig 4a, bottom row). The finding of the DMN and visual networks in the first Granger pair is consistent with this eyes-open, resting dataset [40].

The remaining Granger components tended to cluster into networks that have also been previously reported in the literature. For example, the driving signal of GC2 consisted of the precuneus and posterior cingulate, the other two hubs of the DMN (the pericallosal sulcus borders the posterior cingulate; Fig 4b, top). The driven portion of GC2 was comprised of a contiguous region in the parietal cortex (Fig 4b, bottom), an area belonging to the "sensorimotor" network [40].

The analysis also identified a network in the right frontal region that appeared in the driving portion of GC3 as well as the driven portion of GC4 (Fig 4c, top and Fig 4d, bottom). A set of ROIs in the left temporal cortex was identified in the driving signal of GC4, while the expression of the insular

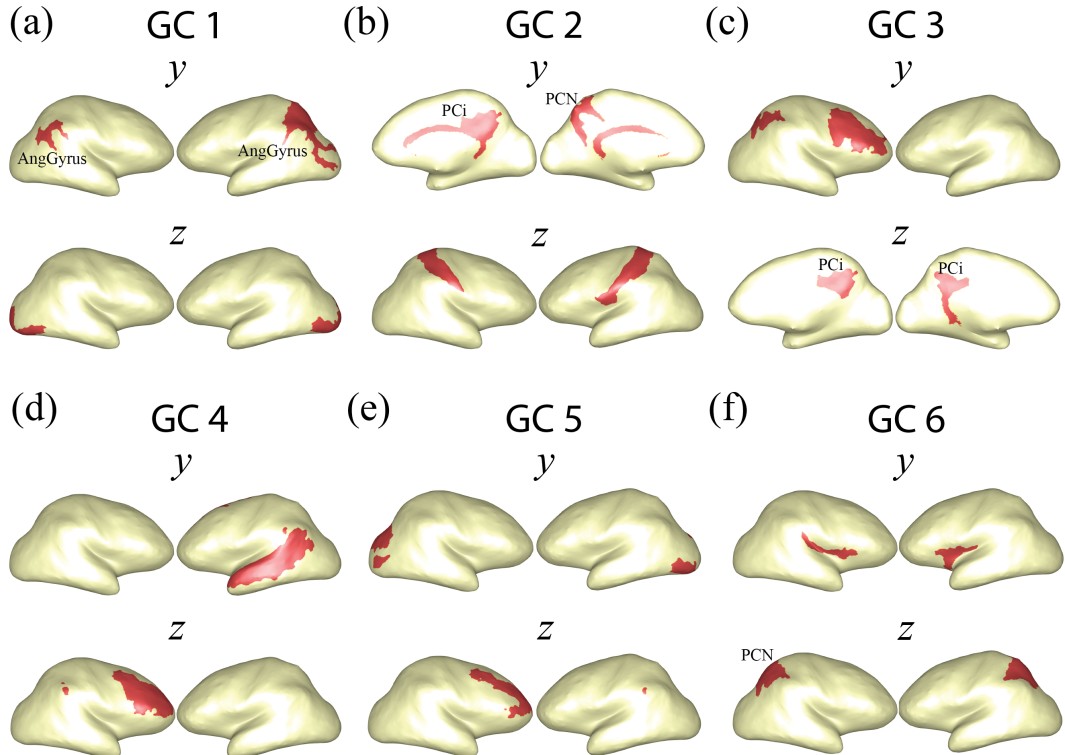

Figure 4: Granger components measured on resting state fMRI data. **(a)** *(Top)* The driving signal of GC1 expressed the left and right angular gyrus, a hub of the DMN. *(Bottom)* The corresponding driven signal identified a visual network including the primary visual cortices. **(b)** The driving portion of GC2 included the other hubs of the DMN, namely the posterior cingulate (PCi) and the precuneus (PCN). Another commonly reported resting-state network, the sensorimotor region, was expressed in the driven component of GC2. **(c)**-**(f)** The brain regions comprising Granger components 3 through 6 were generally clustered into contiguous regions, some of which matched known RSNs (e.g. the "Salience Network" in the driving signal of GC6).

cortex and anterior cingulate in driving GC6 is consistent with another well-known RSN known as the "salience network" [41].

Overall, GCA recovered components whose spatial distribution is consistent with multiple RSNs previously reported in the fMRI literature. Note that the GCs measured here tended to capture RSNs that interact with one another, as opposed to interacting brain regions from the *same* RSN.

## 7  Limitations

There are several limitations of the present work. The proposed method is limited to capturing components that are expressed linearly in the observations. In the high-dimensional case (large $LD$), estimation of the block covariance matrices $\mathbf{\Sigma}_{1:L}$ and $\tilde{\mathbf{\Sigma}}$ is challenging, particularly in the case of low sample sizes. Regularization approaches more sophisticated than the Tikhonov approach employed here may alleviate this challenge [42].

Although the components recovered by GCA are ordered according to the strength of the Granger causal relationship, this work has not proposed a methodology for selecting the number of components $P$ to estimate (or retain). One *post hoc* approach is to perform statistical significance testing on the recovered Granger component pairs to assess the likelihood of a non-spurious relationship. This may be accomplished in a non-parametric fashion by employing the method on surrogate data lacking any Granger causal structure, for example by randomizing the phase spectrum of the observed signals [43]. Furthermore, this work has not considered a methodology for selection of the autoregressive

Table 1: Granger components of resting-state brain activity. Listed are the five ROIs most strongly expressed in the forward models of the driving and driven components of the first $P = 6$ Granger components. Where applicable, the ROIs have been labeled according to previously reported resting-state networks: Visual, Default Mode Network (DMN), Central Executive Network (CEN), Salience Network, Sensorimotor Network.

| $y_1$ | $z_1$ | $y_2$ | $z_2$ |
|---|---|---|---|
| **1** (L) G pariet inf-Angular | (R) Pole occipital | (R) S pericallosal | (R) G postcentral |
| **2** (L) G occipital middle | (L) Pole occipital | (L) G precuneus | (L) G postcentral |
| **3** (L) G parietal sup | (R) G+S occipital inf | (R) G cingul-Post-dorsal | (R) S postcentral |
| **4** (R) G pariet inf-Angular | (L) G+S occipital inf | (L) S pericallosal | (L) G+S subcentral |
| **5** (L) S intrapariet and P tran | (R) S collat transv post | (R) S subparietal | (L) S postcentral |

| $y_3$ | $z_3$ | $y_4$ | $z_4$ |
|---|---|---|---|
| **1** (R) S front inf | (L) S subparietal | (L) S temporal sup | (R) S front inf |
| **2** (R) G front middle | (L) G cingul-Post-ventral | (L) G temporal middle | (R) G front inf-Triangul |
| **3** (R) S precentral-inf-part | (L) G cingul-Post-dorsal | (L) S interm prim-Jensen | (R) S interm prim-Jensen |
| **4** (R) S intrapariet and P tran | (R) G cingul-Post-dorsal | (L) G front sup | (R) G front middle |
| **5** (R) G front inf-Triangul | (R) S subparietal | (L) G temp sup-Lateral | (R) S front middle |

| $y_5$ | $z_5$ | $y_6$ | $z_6$ |
|---|---|---|---|
| **1** (R) G occipital sup | (R) S front middle | (R) S circular insula sup | (L) G parietal sup |
| **2** (L) G occipital sup | (R) G front middle | (L) S circular insula sup | (R) G parietal sup |
| **3** (R) S oc middle and Lunatus | (L) S interm prim-Jensen | (R) Lat Fis-post | (L) S intrapariet and P tran |
| **4** (R) S oc sup and transversal | (R) S orbital lateral | (L) G insular short | (R) G precuneus |
| **5** (L) G+S occipital inf | (R) G+S transv frontopol | (L) G+S cingul-Mid-Ant | (R) S intrapariet and P tran |

model order $L$. Knowledge of the underlying systems was utilized in the two real-world examples here to select the history lengths (i.e., 500 ms for EEG and 4 samples for fMRI).

In defining the GCA objective (3), it has been assumed that the Granger causal relationships between latent variables are unidirectional: if $y$ Granger-causes $z$, then $z$ does not Granger-cause $y$. An extension of GCA that instead maximizes *net* Granger causality [12] and accounts for bidirectional relationships may facilitate the recovery of pairs of sources that exhibit temporal dependencies in both directions.

## 8   Conclusion

This work has proposed an approach to blind source separation that is applicable when the underlying sources are related by a Granger causal relationship (i.e., the past of one signal forecasts the present of another). In contrast to conventional approaches that require statistical independence for identification, the present approach exploits temporal dependencies to recover pairs of components with a directed relationship. A coordinate descent algorithm that decomposes a multivariate dataset into pairs of components, ranked by the strength of the temporal dependency, was then developed and evaluated on simulated and real datasets. Future efforts may extend this line of work to permit non-iterative solutions to the linear GCA problem, or propose modifications that allow one to recover nonlinear temporal dependencies.

The code that was employed to generate the experimental results is available at `https://github.com/dmochow/gca`.

## Acknowledgments

The author would like to thank the anonymous Reviewers for providing many useful comments and critiques that helped to improve the quality of the paper.

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
