# SUPPLEMENTARY MATERIAL
# GRANGER COMPONENTS ANALYSIS: UNSUPERVISED LEARNING OF LATENT TEMPORAL DEPENDENCIES

## Expressions for minimum mean squared errors

In order to obtain a closed-form expression for the GCA objective $J$, expressions for the minimum mean squared error (MMSE) are required for both the reduced and full linear regression models:

$$z(t) = \sum_{l=1}^{L} h_l z(t-l) + \epsilon_r(t) \quad \text{(reduced model)} \tag{1}$$

$$z(t) = \sum_{l=1}^{L} g_{1l} z(t-l) + \sum_{l=1}^{L} g_{2l} y(t-l) + \epsilon_f(t) \quad \text{(full model)} \tag{2}$$

where $h_l$ are the coefficients of the temporal filter predicting the driven signal from its past in the reduced model, $g_{1l}$ are the coefficients of the filter predicting the driven signal from its own past in the full model, and $g_{2l}$ are the temporal filter weights of the filter predicting the driven signal from the past of the driving signal. The regression models can be more compactly written in vector notation as:

$$z(t) = \mathbf{h}^T \mathbf{z}_p(t) + \epsilon_r(t) \tag{3}$$

$$z(t) = \begin{bmatrix} \mathbf{g}_1 & \mathbf{g}_2 \end{bmatrix} \begin{bmatrix} \mathbf{z}_p(t) \\ \mathbf{y}_p(t) \end{bmatrix} + \epsilon_f(t). \tag{4}$$

The coefficients of the filter $\mathbf{h}$ that minimizes the residual in the reduced model are given by the Wiener filter [1]:

$$\mathbf{h} = \mathbf{Q}^{-1}\mathbf{q}, \tag{5}$$

where $\mathbf{q} = E\{z(t)\mathbf{z}_p(t)\}$ is the covariance vector between the desired signal $z$ and its own past, whose $l$th element is given by $E\{z(t)z(t-l)\}$, and where $\mathbf{Q} = E\{\mathbf{z}_p(t)\mathbf{z}_p^T(t)\}$ is the covariance matrix of $\mathbf{z}_p(t)$, where the element at row $i$ and column $j$ is given by $E\{z(t-i)z(t-j)\} = E\{z(t)z(t-j+i)\}$ under the assumption of wide-sense stationary observation data. By substituting (5) into (3) and solving for the residual power, the corresponding MMSE follows as:

$$\Phi_r = \langle \epsilon_{zy}^2 \rangle = \sigma_z^2 - \mathbf{q}^T \mathbf{Q}^{-1} \mathbf{q}. \tag{6}$$

It is required to express $\sigma_z$, $\mathbf{q}$, and $\mathbf{Q}$ in terms of the projection vector $\mathbf{v}$ and the statistics of the observed data $\mathbf{x}(t)$. The power of the desired signal $z$ in the regression models is given by:

$$\sigma_z^2 = E\{z^2(t)\} = \mathbf{v}^T \mathbf{\Sigma}(0)\mathbf{v}, \tag{7}$$

where $\mathbf{\Sigma}(\tau) = E\left\{\mathbf{x}(t)\mathbf{x}^T(t-\tau)\right\}$ is the lagged covariance matrix of the observed data. Substitution of $z(t) = \mathbf{v}^T\mathbf{x}(t)$ into the expression for $\mathbf{q}$ leads to:

$$
\begin{aligned}
\mathbf{q} &= E\left\{z(t)\mathbf{z}_p(t)\right\} \\
&= \begin{pmatrix} E\left\{z(t)z(t-1)\right\} \\ E\left\{z(t)z(t-2)\right\} \\ \vdots \\ E\left\{z(t)z(t-L)\right\} \end{pmatrix} \\
&= \begin{pmatrix} \mathbf{v}^T\mathbf{\Sigma}(1)\mathbf{v} \\ \mathbf{v}^T\mathbf{\Sigma}(2)\mathbf{v} \\ \vdots \\ \mathbf{v}^T\mathbf{\Sigma}(L)\mathbf{v} \end{pmatrix} \\
&= \begin{pmatrix} \mathbf{v}^T & & & \\ & \mathbf{v}^T & & \\ & & \ddots & \\ & & & \mathbf{v}^T \end{pmatrix} \begin{pmatrix} \mathbf{\Sigma}(1) & & & \\ & \mathbf{\Sigma}(2) & & \\ & & \ddots & \\ & & & \mathbf{\Sigma}(L) \end{pmatrix} \begin{pmatrix} \mathbf{v} \\ \mathbf{v} \\ \vdots \\ \mathbf{v} \end{pmatrix} . \quad (8)
\end{aligned}
$$

By utilizing the Kronecker product $\otimes$, one can write (8) as the following matrix product:

$$
\mathbf{q} = \left(\mathbf{I}_L \otimes \mathbf{v}^T\right)\mathbf{\Sigma}_{1:L}\left(\mathbf{1}_L \otimes \mathbf{v}\right), \quad (9)
$$

where

$$
\mathbf{\Sigma}_{1:L} = \begin{pmatrix} \mathbf{\Sigma}(1) & \mathbf{0} & \dots & \mathbf{0} \\ \mathbf{0} & \mathbf{\Sigma}(2) & \dots & \mathbf{0} \\ \vdots & \mathbf{0} & \ddots & \vdots \\ \mathbf{0} & \dots & \dots & \mathbf{\Sigma}(L) \end{pmatrix}
$$

is an $LD$-by-$LD$ block diagonal covariance matrix, $\mathbf{1}_L$ is a vector of all ones, and $\mathbf{I}_L$ is the $L$-by-$L$ identity matrix.

Similarly, the covariance matrix $\mathbf{Q}$ can be written as:

$$
\begin{aligned}
\mathbf{Q} &= E\left\{\mathbf{z}_p(t)\mathbf{z}_p^T(t)\right\} \\
&= \begin{bmatrix} E\left\{z^2(t-1)\right\} & E\left\{z(t-1)z(t-2)\right\} & \dots & E\left\{z(t-1)z(t-L)\right\} \\ E\left\{z(t-2)z(t-1)\right\} & E\left\{z^2(t-2)\right\} & \dots & E\left\{z(t-2)z(t-L)\right\} \\ \vdots & & \ddots & \vdots \\ E\left\{z(t-L)z(t-1)\right\} & & \dots & E\left\{z^2(t-L)\right\} \end{bmatrix} \\
&= \begin{bmatrix} \mathbf{v}^T\mathbf{\Sigma}(0)\mathbf{v} & \mathbf{v}^T\mathbf{\Sigma}(1)\mathbf{v} & \dots & \mathbf{v}^T\mathbf{\Sigma}(L-1)\mathbf{v} \\ \mathbf{v}^T\mathbf{\Sigma}(-1)\mathbf{v} & \mathbf{v}^T\mathbf{\Sigma}(0)\mathbf{v} & \dots & \mathbf{v}^T\mathbf{\Sigma}(L-2)\mathbf{v}] \\ \vdots & & \ddots & \vdots \\ \mathbf{v}^T\mathbf{\Sigma}(-L+1)\mathbf{v} & & \dots & \mathbf{v}^T\mathbf{\Sigma}(0)\mathbf{v} \end{bmatrix} \\
&= \left(\mathbf{I}_L \otimes \mathbf{v}\right)^T \tilde{\mathbf{\Sigma}}\left(\mathbf{I}_L \otimes \mathbf{v}\right),
\end{aligned}
$$

where

$$
\tilde{\mathbf{\Sigma}} = \begin{pmatrix} \mathbf{\Sigma}(0) & \mathbf{\Sigma}(-1) & \dots & \mathbf{\Sigma}(-L+1) \\ \mathbf{\Sigma}(1) & \mathbf{\Sigma}(0) & \dots & \mathbf{\Sigma}(-L+2) \\ \vdots & \mathbf{\Sigma}(1) & \ddots & \vdots \\ \mathbf{\Sigma}(L-1) & \dots & \dots & \mathbf{\Sigma}(0) \end{pmatrix}
$$

is an $LD$-by-$LD$ block Toeplitz matrix. The elements of matrix $\mathbf{Q}$ can now be differentiated with respect to the elements of the projection vector $\mathbf{v}$.

A similar development can be performed to derive at closed-form expressions for the covariance vector $\mathbf{r}$ and covariance matrix $\mathbf{R}$. The Wiener filter of the full regression model (3) is given by:

$$
\begin{aligned}
\mathbf{g} &= \begin{pmatrix} \mathbf{g}_1 \\ \mathbf{g}_2 \end{pmatrix} \\
&= \mathbf{R}^{-1}\mathbf{r}, \quad (10)
\end{aligned}
$$

and the corresponding MMSE is given by:

$$\Phi_f = \left\langle \epsilon_{zy}^2 \right\rangle = \sigma_z^2 - \mathbf{r}^T \mathbf{R}^{-1} \mathbf{r}. \tag{11}$$

Substituting $z(t) = \mathbf{v}^T \mathbf{x}(t)$ and $y(t) = \mathbf{w}^T \mathbf{x}(t)$ into the definition of $\mathbf{r}$ yields:

$$
\begin{aligned}
\mathbf{r} &= E\left\{ z(t) \begin{pmatrix} \mathbf{z}_p(t) \\ \mathbf{y}_p(t) \end{pmatrix} \right\} \\[2mm]
&= \begin{pmatrix}
E\{z(t)z(t-1)\} \\
E\{z(t)z(t-2)\} \\
\vdots \\
E\{z(t)z(t-L)\} \\
E\{z(t)y(t-1)\} \\
E\{z(t)y(t-2)\} \\
\vdots \\
E\{z(t)y(t-L)\}
\end{pmatrix} \\[2mm]
&= \begin{pmatrix}
\mathbf{v}^T \mathbf{\Sigma}(1)\mathbf{v} \\
\mathbf{v}^T \mathbf{\Sigma}(2)\mathbf{v} \\
\vdots \\
\mathbf{v}^T \mathbf{\Sigma}(L)\mathbf{v} \\
\mathbf{v}^T \mathbf{\Sigma}(1)\mathbf{w} \\
\mathbf{v}^T \mathbf{\Sigma}(2)\mathbf{w} \\
\vdots \\
\mathbf{v}^T \mathbf{\Sigma}(L)\mathbf{w}
\end{pmatrix}.
\end{aligned} \tag{12}
$$

It is straightforward to verify that (12) can be factored according to:

$$\mathbf{r} = \left(\mathbf{I}_{2L} \otimes \mathbf{v}^T\right) \left(\mathbf{I}_2 \otimes \mathbf{\Sigma}_{1:L}\right) \begin{pmatrix} \mathbf{1}_L \otimes \mathbf{v} \\ \mathbf{1}_L \otimes \mathbf{w} \end{pmatrix}. \tag{13}$$

Finally, the explicit expression for covariance matrix $\mathbf{R}$ is given by:

$$
\mathbf{R} = E\left\{ \begin{pmatrix} \mathbf{z}_p(t) \\ \mathbf{y}_p(t) \end{pmatrix} \begin{pmatrix} \mathbf{z}_p(t) \\ \mathbf{y}_p(t) \end{pmatrix}^T \right\}
$$

$$
= \begin{pmatrix}
\mathbf{v}^T\mathbf{\Sigma}(0)\mathbf{v} & \mathbf{v}^T\mathbf{\Sigma}(1)\mathbf{v} & \dots & \mathbf{v}^T\mathbf{\Sigma}(L-1)\mathbf{v} & \mathbf{v}^T\mathbf{\Sigma}(0)\mathbf{w} & \mathbf{v}^T\mathbf{\Sigma}(1)\mathbf{w} & \dots & \mathbf{v}^T\mathbf{\Sigma}(L-1)\mathbf{w} \\
\mathbf{v}^T\mathbf{\Sigma}(-1)\mathbf{v} & \mathbf{v}^T\mathbf{\Sigma}(0)\mathbf{v} & \dots & \mathbf{v}^T\mathbf{\Sigma}(L-2)\mathbf{v} & \mathbf{v}^T\mathbf{\Sigma}(-1)\mathbf{w} & \mathbf{v}^T\mathbf{\Sigma}(0)\mathbf{w} & \dots & \mathbf{v}^T\mathbf{\Sigma}(L-2)\mathbf{w} \\
\vdots & & \ddots & \vdots & & & & \\
\mathbf{v}^T\mathbf{\Sigma}(-L+1)\mathbf{v} & & \dots & \mathbf{v}^T\mathbf{\Sigma}(0)\mathbf{v} & \mathbf{v}^T\mathbf{\Sigma}(-L+1)\mathbf{w} & & \dots & \mathbf{v}^T\mathbf{\Sigma}(0)\mathbf{w} \\
\mathbf{w}^T\mathbf{\Sigma}(0)\mathbf{v} & \mathbf{w}^T\mathbf{\Sigma}(1)\mathbf{v} & \dots & \mathbf{w}^T\mathbf{\Sigma}(L-1)\mathbf{v} & \mathbf{w}^T\mathbf{\Sigma}(0)\mathbf{w} & \mathbf{w}^T\mathbf{\Sigma}(1)\mathbf{w} & \dots & \mathbf{w}^T\mathbf{\Sigma}(L-1)\mathbf{w} \\
\mathbf{w}^T\mathbf{\Sigma}(-1)\mathbf{w} & \mathbf{w}^T\mathbf{\Sigma}(0)\mathbf{v} & \dots & \mathbf{w}^T\mathbf{\Sigma}(L-2)\mathbf{v} & \mathbf{w}^T\mathbf{\Sigma}(-1)\mathbf{w} & \mathbf{w}^T\mathbf{\Sigma}(0)\mathbf{w} & \dots & \mathbf{w}^T\mathbf{\Sigma}(L-2)\mathbf{w} \\
\vdots & & \ddots & \vdots & & & & \\
\mathbf{w}^T\mathbf{\Sigma}(-L+1)\mathbf{v} & & \dots & \mathbf{w}^T\mathbf{\Sigma}(0)\mathbf{v} & \mathbf{w}^T\mathbf{\Sigma}(-L+1)\mathbf{w} & & \dots & \mathbf{w}^T\mathbf{\Sigma}(0)\mathbf{w}
\end{pmatrix},
$$

which can be factored according to:

$$\mathbf{R} = \begin{pmatrix} \mathbf{1}_{\frac{T}{2}} \otimes \mathbf{I}_L \otimes \mathbf{v}^T \\ \mathbf{1}_{\frac{T}{2}} \otimes \mathbf{I}_L \otimes \mathbf{w}^T \end{pmatrix} \left(\mathbf{I}_2 \otimes \tilde{\mathbf{\Sigma}}\right) \begin{pmatrix} \mathbf{I}_L \otimes \mathbf{v} & \mathbf{0} \\ \mathbf{0} & \mathbf{I}_L \otimes \mathbf{w} \end{pmatrix}. \tag{14}$$

## Gradient of GCA objective function

The goal is to differentiate the objective function:

$$
\begin{aligned}
J(\mathbf{w}, \mathbf{v}) &= \log \frac{\Phi_f}{\Phi_r} \\
&= \log\left(\sigma_z^2 - \mathbf{r}^T \mathbf{R}^{-1} \mathbf{r}\right) - \log\left(\sigma_z^2 - \mathbf{q}^T \mathbf{Q}^{-1} \mathbf{q}\right)
\end{aligned}
\tag{15}
$$

with respect to the projection vectors $\mathbf{v}$ and $\mathbf{w}$. The derivation below relies on the chain rule, and involves the differentiation of matrices $\mathbf{R}$ and $\mathbf{Q}$ with respect to vectors $\mathbf{v}$ and $\mathbf{w}$. The reader is referred to Magnus and Neudecker [2] for an excellent treatment of matrix differentials, which is utilized here.

Taking differentials of both sides of (15) yields:

$$
\begin{aligned}
dJ &= \frac{\Phi_f d\Phi_r - \Phi_r d\Phi_f}{\Phi_f \Phi_r} \\
&= \frac{1}{\Phi_r}\left(d\sigma_z^2 - d\left(\mathbf{q}^T\mathbf{Q}^{-1}\mathbf{q}\right)\right) - \frac{1}{\Phi_f}\left(d\sigma_z^2 - d\left(\mathbf{r}^T\mathbf{R}^{-1}\mathbf{r}\right)\right) \\
&= \frac{1}{\Phi_r}\left(d\sigma_z^2 - d\mathbf{q}^T\mathbf{Q}^{-1}\mathbf{q} - \mathbf{q}^T d\mathbf{Q}^{-1}\mathbf{q} - \mathbf{q}^T\mathbf{Q}^{-1}d\mathbf{q}\right) - \frac{1}{\Phi_f}\left(d\sigma_z^2 - d\mathbf{r}^T\mathbf{R}^{-1}\mathbf{r} - \mathbf{r}^T d\mathbf{R}^{-1}\mathbf{r} - \mathbf{r}^T\mathbf{R}^{-1}d\mathbf{r}\right) \\
&= \frac{1}{\Phi_r}\left(d\sigma_z^2 - 2\mathbf{q}^T\mathbf{Q}^{-1}d\mathbf{q} + \mathbf{q}^T\mathbf{Q}^{-1}d\mathbf{Q}\mathbf{Q}^{-1}\mathbf{q}\right) - \frac{1}{\Phi_f}\left(d\sigma_z^2 - 2\mathbf{r}^T\mathbf{R}^{-1}d\mathbf{r} + \mathbf{r}^T\mathbf{R}^{-1}d\mathbf{R}\mathbf{R}^{-1}\mathbf{r}\right) \\
&= \frac{1}{\Phi_r}\left(d\sigma_z^2 - 2\mathbf{q}^T\mathbf{Q}^{-1}d\mathbf{q} + \operatorname{tr}\mathbf{q}^T\mathbf{Q}^{-1}d\mathbf{Q}\mathbf{Q}^{-1}\mathbf{q}\right) - \frac{1}{\Phi_f}\left(d\sigma_z^2 - 2\mathbf{r}^T\mathbf{R}^{-1}d\mathbf{r} + \operatorname{tr}\mathbf{r}^T\mathbf{R}^{-1}d\mathbf{R}\mathbf{R}^{-1}\mathbf{r}\right) \\
&= \frac{1}{\Phi_r}\left(d\sigma_z^2 - 2\mathbf{q}^T\mathbf{Q}^{-1}d\mathbf{q} + \operatorname{tr}\mathbf{Q}^{-1}\mathbf{q}\mathbf{q}^T\mathbf{Q}^{-1}d\mathbf{Q}\right) - \frac{1}{\Phi_f}\left(d\sigma_z^2 - 2\mathbf{r}^T\mathbf{R}^{-1}d\mathbf{r} + \operatorname{tr}\mathbf{R}^{-1}\mathbf{r}\mathbf{r}^T\mathbf{R}^{-1}d\mathbf{R}\right) \\
&= \frac{1}{\Phi_r}\left(d\sigma_z^2 - 2\mathbf{q}^T\mathbf{Q}^{-1}d\mathbf{q} + \operatorname{vec}\left(\mathbf{Q}^{-1}\mathbf{q}\mathbf{q}^T\mathbf{Q}^{-1}\right)^T \operatorname{vec} d\mathbf{Q}\right) \\
&\quad - \frac{1}{\Phi_f}\left(d\sigma_z^2 - 2\mathbf{r}^T\mathbf{R}^{-1}d\mathbf{r} + \operatorname{vec}\left(\mathbf{R}^{-1}\mathbf{r}\mathbf{r}^T\mathbf{R}^{-1}\right)^T \operatorname{vec} d\mathbf{R}\right) \\
&= \frac{1}{\Phi_r}\left(2\mathbf{v}^T\mathbf{\Sigma}(0)d\mathbf{v} - 2\mathbf{q}^T\mathbf{Q}^{-1}\mathbf{J}_q d\mathbf{v} + \operatorname{vec}\left(\mathbf{Q}^{-1}\mathbf{q}\mathbf{q}^T\mathbf{Q}^{-1}\right)^T \mathbf{J}_Q d\mathbf{v}\right) \\
&\quad - \frac{1}{\Phi_f}\left(2\mathbf{v}^T\mathbf{\Sigma}(0)d\mathbf{v} - 2\mathbf{r}^T\mathbf{R}^{-1}\mathbf{J}_r\begin{pmatrix} d\mathbf{v} \\ d\mathbf{w} \end{pmatrix} + \operatorname{vec}\left(\mathbf{R}^{-1}\mathbf{r}\mathbf{r}^T\mathbf{R}^{-1}\right)^T \mathbf{J}_R\begin{pmatrix} d\mathbf{v} \\ d\mathbf{w} \end{pmatrix}\right),
\end{aligned}
\tag{16}
$$

where $\operatorname{tr}$ is the matrix trace operator, $\operatorname{vec}$ is an operator that transforms a matrix into a column vector by stacking the columns, and where the following Jacobian matrices have been defined:

$$
\begin{aligned}
d\mathbf{q} &= \mathbf{J}_q d\mathbf{v} \\
\operatorname{vec} d\mathbf{Q} &= \mathbf{J}_Q d\mathbf{v} \\
d\mathbf{r} &= \mathbf{J}_r\begin{pmatrix} d\mathbf{v} \\ d\mathbf{w} \end{pmatrix} \\
\operatorname{vec} d\mathbf{R} &= \mathbf{J}_R\begin{pmatrix} d\mathbf{v} \\ d\mathbf{w} \end{pmatrix}.
\end{aligned}
$$

Once closed-form expressions for these Jacobians are derived, it remains to substitute those expressions into (16).

### Differential of $\mathbf{q}$

The covariance vector $\mathbf{q}$ is defined by:

$$
\mathbf{q} = \left(\mathbf{I}_L \otimes \mathbf{v}^T\right)\mathbf{\Sigma}_{1:L}\left(\mathbf{1}_L \otimes \mathbf{v}\right).
\tag{17}
$$

The goal is to arrive at an expression of the form:

$$
d\mathbf{q} = \mathbf{J}_q d\mathbf{v},
\tag{18}
$$

where $\mathbf{J}_q$ is the Jacobian matrix that one seeks to identify. The following identity (often termed the "vec" rule) will prove useful throughout:

$$\mathrm{vec}(\mathbf{ABC}) = \left(\mathbf{C}^T \otimes \mathbf{A}\right) \mathrm{vec}\left(\mathbf{B}\right),$$

where matrices $\mathbf{A}$, $\mathbf{B}$, and $\mathbf{C}$ are defined such that the conventional matrix product $\mathbf{ABC}$ is valid. Taking differentials of both sides of (17) and applying the vec rule, one obtains:

$$d\mathbf{q} \quad = \quad \left(\left(\mathbf{1}_L^T \otimes \mathbf{v}^T\right) \boldsymbol{\Sigma}_{1:L}^T \otimes \mathbf{I}_L\right) d\,\mathrm{vec}\left(\mathbf{I}_L \otimes \mathbf{v}^T\right) + \left(\mathbf{I}_L \otimes \mathbf{v}\right)^T \boldsymbol{\Sigma}_{1:L}\, d\,\mathrm{vec}\left(\mathbf{1}_L \otimes \mathbf{v}\right). \tag{19}$$

The differentials on the right-hand side may be written as [2]:

$$d\,\mathrm{vec}\left(\mathbf{1}_L \otimes \mathbf{v}\right) = \left(\mathbf{1}_L \otimes \mathbf{I}_D\right) d\mathbf{v}, \tag{20}$$

and

$$d\,\mathrm{vec}\left(\mathbf{I}_L \otimes \mathbf{v}^T\right) = \left(\mathbf{I}_L \otimes \mathbf{K}_{DL}\right) \left(\mathrm{vec}\,\mathbf{I}_L \otimes \mathbf{I}_D\right) d\mathbf{v}, \tag{21}$$

where $\mathbf{K}_{DL}$ is a commutation matrix satisfying:

$$\mathbf{K}_{mn}\mathrm{vec}\left(\mathbf{D}\right) = \mathrm{vec}\left(\mathbf{D}^T\right)$$

for $m$-by-$n$ matrix $\mathbf{D}$. In deriving (20) and (21), the property on p. 206 of Magnus and Neudecker [2] has been invoked to move the differential outside of the vec operator. Substituting (20) and (21) into (19), one obtains the required result:

$$d\mathbf{q} = \left[\left(\left(\mathbf{1}_L^T \otimes \mathbf{v}^T\right) \boldsymbol{\Sigma}_{1:L}^T \otimes \mathbf{I}_L\right) \left(\mathbf{I}_L \otimes \mathbf{K}_{DL}\right) \left(\mathrm{vec}\,\mathbf{I}_L \otimes \mathbf{I}_D\right) + \left(\mathbf{I}_L \otimes \mathbf{v}\right)^T \boldsymbol{\Sigma}_{1:L} \left(\mathbf{1}_L \otimes \mathbf{I}_D\right)\right] d\mathbf{v},$$

where the Jacobian is identified as:

$$\mathbf{J}_q = \left(\left(\mathbf{1}_L^T \otimes \mathbf{v}^T\right) \boldsymbol{\Sigma}_{1:L}^T \otimes \mathbf{I}_L\right) \left(\mathbf{I}_L \otimes \mathbf{K}_{DL}\right) \left(\mathrm{vec}\,\mathbf{I}_L \otimes \mathbf{I}_D\right) + \left(\mathbf{I}_L \otimes \mathbf{v}\right)^T \boldsymbol{\Sigma}_{1:L} \left(\mathbf{1}_L \otimes \mathbf{I}_D\right).$$

**Differential of Q**

The $L$-by-$L$ covariance matrix of predictors in the reduced regression model is given by:

$$\mathbf{Q} = \left(\mathbf{I}_L \otimes \mathbf{v}\right)^T \tilde{\boldsymbol{\Sigma}} \left(\mathbf{I}_L \otimes \mathbf{v}\right).$$

The differential follows as:

$$d\mathbf{Q} = \left(\mathbf{I}_L \otimes d\mathbf{v}\right)^T \tilde{\boldsymbol{\Sigma}} \left(\mathbf{I}_L \otimes \mathbf{v}\right) + \left(\mathbf{I}_L \otimes \mathbf{v}\right)^T \tilde{\boldsymbol{\Sigma}} \left(\mathbf{I}_L \otimes d\mathbf{v}\right). \tag{22}$$

By vectorizing both sides of (22) and applying the vec rule to both terms on the right-hand side, one obtains:

$$\mathrm{vec}\, d\mathbf{Q} = \left[\left(\tilde{\boldsymbol{\Sigma}} \left(\mathbf{I}_L \otimes \mathbf{v}\right)\right)^T \otimes \mathbf{I}_L\right] \mathrm{vec}\left(\mathbf{I}_L \otimes d\mathbf{v}^T\right) + \left(\mathbf{I}_L \otimes \left(\mathbf{I}_L \otimes \mathbf{v}^T\right) \tilde{\boldsymbol{\Sigma}}\right) \mathrm{vec}\left(\mathbf{I}_L \otimes d\mathbf{v}\right).$$

Again using the property at the bottom of page 206 in Magnus and Neudecker [2], one obtains the following expression:

$$\mathrm{vec}\, d\mathbf{Q} \quad = \quad \left[\left(\mathbf{I}_L \otimes \mathbf{v}^T\right) \tilde{\boldsymbol{\Sigma}}^T \otimes \mathbf{I}_L\right] \left(\mathbf{I}_L \otimes \mathbf{K}_{DL}\right) \left(\mathrm{vec}\,\mathbf{I}_L \otimes \mathbf{I}_D\right) d\mathbf{v} +$$

$$\left(\mathbf{I}_L \otimes \left(\mathbf{I}_L \otimes \mathbf{v}^T\right) \tilde{\boldsymbol{\Sigma}}\right) \left(\mathbf{I}_L \otimes \mathbf{I}_L \otimes \mathbf{I}_D\right) \left(\mathrm{vec}\,\mathbf{I}_L \otimes \mathbf{I}_D\right) d\mathbf{v},$$

from which the Jacobian is identified as:

$$\mathbf{J}_Q \quad = \quad \left[\left(\mathbf{I}_L \otimes \mathbf{v}^T\right) \tilde{\boldsymbol{\Sigma}}^T \otimes \mathbf{I}_L\right] \left(\mathbf{I}_L \otimes \mathbf{K}_{DL}\right) \left(\mathrm{vec}\,\mathbf{I}_L \otimes \mathbf{I}_D\right) + \left(\mathbf{I}_L \otimes \left(\mathbf{I}_L \otimes \mathbf{v}^T\right) \tilde{\boldsymbol{\Sigma}}\right) \left(\mathbf{I}_L \otimes \mathbf{I}_L \otimes \mathbf{I}_D\right) \left(\mathrm{vec}\,\mathbf{I}_L \otimes \mathbf{I}_D\right).$$

**Differential of r**

The $2L$-dimensional covariance vector in the full regression model is given by:

$$\mathbf{r} = \left(\mathbf{I}_{2L} \otimes \mathbf{v}^T\right) \left(\mathbf{I}_2 \otimes \boldsymbol{\Sigma}_{1:L}\right) \begin{pmatrix} \mathbf{1}_L \otimes \mathbf{v} \\ \mathbf{1}_L \otimes \mathbf{w} \end{pmatrix}. \tag{23}$$

Taking differentials of (23), one obtains:

$$d\mathbf{r} = \left(\mathbf{I}_{2L} \otimes d\mathbf{v}^T\right) \left(\mathbf{I}_2 \otimes \boldsymbol{\Sigma}_{1:L}\right) \begin{pmatrix} \mathbf{1}_L \otimes \mathbf{v} \\ \mathbf{1}_L \otimes \mathbf{w} \end{pmatrix} + \left(\mathbf{I}_{2L} \otimes \mathbf{v}^T\right) \left(\mathbf{I}_2 \otimes \boldsymbol{\Sigma}_{1:L}\right) \begin{pmatrix} \mathbf{1}_L \otimes d\mathbf{v} \\ \mathbf{1}_L \otimes d\mathbf{w} \end{pmatrix}. \tag{24}$$

Applying the vec operator to both sides of (24) yields:

$$\text{vec } d\mathbf{r} = \text{vec}\left(\left(\mathbf{I}_{2L} \otimes d\mathbf{v}^T\right)\left(\mathbf{I}_2 \otimes \boldsymbol{\Sigma}_{1:L}\right)\begin{pmatrix} \mathbf{1}_L \otimes \mathbf{v} \\ \mathbf{1}_L \otimes \mathbf{w} \end{pmatrix}\right) + \text{vec}\left(\left(\mathbf{I}_{2L} \otimes \mathbf{v}^T\right)\left(\mathbf{I}_2 \otimes \boldsymbol{\Sigma}_{1:L}\right)\begin{pmatrix} \mathbf{1}_L \otimes d\mathbf{v} \\ \mathbf{1}_L \otimes d\mathbf{w} \end{pmatrix}\right). \quad (25)$$

The vec rule can now be applied to the right hand side of (25):

$$d\mathbf{r} = \left(\left(\begin{pmatrix} \mathbf{1}_L \otimes \mathbf{v} \\ \mathbf{1}_L \otimes \mathbf{w} \end{pmatrix}^T \left(\mathbf{I}_2 \otimes \boldsymbol{\Sigma}_{1:L}\right)^T \otimes \mathbf{I}_{2L}\right)\text{vec}\left(\mathbf{I}_{2L} \otimes d\mathbf{v}^T\right) + \left(\mathbf{I}_{2L} \otimes \mathbf{v}^T\right)\left(\mathbf{I}_2 \otimes \boldsymbol{\Sigma}_{1:L}\right)\begin{pmatrix} \text{vec } \left(\mathbf{1}_L \otimes d\mathbf{v}\right) \\ \text{vec } \left(\mathbf{1}_L \otimes d\mathbf{w}\right) \end{pmatrix}\right).$$

Applying the property on page 206 of Magnus and Neudecker [2], the resulting expression may be written as:

$$d\mathbf{r} = \left(\begin{pmatrix} \mathbf{1}_L \otimes \mathbf{v} \\ \mathbf{1}_L \otimes \mathbf{w} \end{pmatrix}^T \left(\mathbf{I}_2 \otimes \boldsymbol{\Sigma}_{1:L}\right)^T \otimes \mathbf{I}_{2L}\right)\left(\mathbf{I}_{2L} \otimes \mathbf{K}_{D,2L}\right)\left(\text{vec } \mathbf{I}_{2L} \otimes \mathbf{I}_D\right) d\mathbf{v} +$$

$$\left(\mathbf{I}_{2L} \otimes \mathbf{v}^T\right)\left(\mathbf{I}_2 \otimes \boldsymbol{\Sigma}_{1:L}\right)\begin{pmatrix} \left(\text{vec } \mathbf{1}_L \otimes \mathbf{I}_D\right) d\mathbf{v} \\ \left(\text{vec } \mathbf{1}_L \otimes \mathbf{I}_D\right) d\mathbf{w} \end{pmatrix},$$

which can then be expressed as:

$$d\mathbf{r} = \left(\begin{pmatrix} \mathbf{1}_L \otimes \mathbf{v} \\ \mathbf{1}_L \otimes \mathbf{w} \end{pmatrix}^T \left(\mathbf{I}_2 \otimes \boldsymbol{\Sigma}_{1:L}\right)^T \otimes \mathbf{I}_{2L}\right)\left(\mathbf{I}_{2L} \otimes \mathbf{K}_{D,2L}\right)\left(\text{vec } \mathbf{I}_{2L} \otimes \mathbf{I}_D\right) d\mathbf{v} +$$

$$\left(\mathbf{I}_{2L} \otimes \mathbf{v}^T\right)\left(\mathbf{I}_2 \otimes \boldsymbol{\Sigma}_{1:L}\right)\begin{pmatrix} \mathbf{1}_L \otimes \mathbf{I}_D & \mathbf{0} \\ \mathbf{0} & \mathbf{1}_L \otimes \mathbf{I}_D \end{pmatrix}\begin{pmatrix} d\mathbf{v} \\ d\mathbf{w} \end{pmatrix}.$$

The Jacobian of $\mathbf{r}$ with respect to $\mathbf{v}$ and $\mathbf{w}$ can now be identified:

$$\mathbf{J}_r = \left[\left(\begin{pmatrix} \mathbf{1}_L \otimes \mathbf{v} \\ \mathbf{1}_L \otimes \mathbf{w} \end{pmatrix}^T \left(\mathbf{I}_2 \otimes \boldsymbol{\Sigma}_{1:L}\right)^T \otimes \mathbf{I}_{2L}\right)\left(\mathbf{I}_{2L} \otimes \mathbf{K}_{D,2L}\right)\left(\text{vec } \mathbf{I}_{2L} \otimes \mathbf{I}_D\right) \quad \mathbf{0}\right] +$$

$$\left(\mathbf{I}_{2L} \otimes \mathbf{v}^T\right)\left(\mathbf{I}_2 \otimes \boldsymbol{\Sigma}_{1:L}\right)\begin{pmatrix} \mathbf{1}_L \otimes \mathbf{I}_D & \mathbf{0} \\ \mathbf{0} & \mathbf{1}_L \otimes \mathbf{I}_D \end{pmatrix}.$$

**Differential of $\mathbf{R}$**

The covariance matrix of predictors in the full regression model is given by the $2L$-by-$2L$ matrix:

$$\mathbf{R} = \begin{pmatrix} \mathbf{1}_{\frac{T}{2}} \otimes \mathbf{I}_L \otimes \mathbf{v}^T \\ \mathbf{1}_{\frac{T}{2}} \otimes \mathbf{I}_L \otimes \mathbf{w}^T \end{pmatrix}\left(\mathbf{I}_2 \otimes \tilde{\boldsymbol{\Sigma}}\right)\begin{pmatrix} \mathbf{I}_L \otimes \mathbf{v} & \mathbf{0} \\ \mathbf{0} & \mathbf{I}_L \otimes \mathbf{w} \end{pmatrix}.$$

By following the same development as above, the differential of the elements of $\mathbf{R}$ is given by:

$$\text{vec } d\mathbf{R}$$
$$= \left[\left(\begin{matrix} \mathbf{I}_L \otimes \mathbf{v}^T & \mathbf{0} \\ \mathbf{0} & \mathbf{I}_L \otimes \mathbf{w}^T \end{matrix}\right)\left(\mathbf{I}_2 \otimes \tilde{\boldsymbol{\Sigma}}^T\right) \otimes \mathbf{I}_{2L}\right]\left(\mathbf{K}_{2,2LD} \otimes \mathbf{I}_L\right)^{-1}\left\{\mathbf{I}_2 \otimes \left[\left(\mathbf{I}_{2L} \otimes \mathbf{K}_{DL}\right)\left(\text{vec } \left(\mathbf{1}_{\frac{T}{2}} \otimes \mathbf{I}_L\right) \otimes \mathbf{I}_D\right)\right]\right\}\begin{pmatrix} d\mathbf{v} \\ d\mathbf{w} \end{pmatrix}$$
$$+ \left[\mathbf{I}_{2L} \otimes \left(\begin{matrix} \mathbf{1}_{\frac{T}{2}}^T \otimes \mathbf{I}_L \otimes \mathbf{v}^T \\ \mathbf{1}_{\frac{T}{2}}^T \otimes \mathbf{I}_L \otimes \mathbf{w}^T \end{matrix}\right)\left(\mathbf{I}_2 \otimes \tilde{\boldsymbol{\Sigma}}\right)\right]\left(\mathbf{I}_2 \otimes \mathbf{K}_{2,L} \otimes \mathbf{I}_{LD}\right)^{-1}\left[\mathbf{I}_{4,2} \otimes \left(\mathbf{I}_L \otimes \mathbf{I}_L \otimes \mathbf{I}_D\right)\left(\text{vec } \mathbf{I}_L \otimes \mathbf{I}_D\right)\right]\begin{pmatrix} d\mathbf{v} \\ d\mathbf{w} \end{pmatrix}, \quad (26)$$

where $\mathbf{I}_{4,2}$ is a 4 by 2 matrix with ones at row 1, column 1 and at row 4, column 2, and zeros elsewhere. Thus, the Jacobian matrix $\mathbf{J}_R$ is identified as:

$$\mathbf{J}_R$$
$$= \left[\left(\begin{matrix} \mathbf{I}_L \otimes \mathbf{v}^T & \mathbf{0} \\ \mathbf{0} & \mathbf{I}_L \otimes \mathbf{w}^T \end{matrix}\right)\left(\mathbf{I}_2 \otimes \tilde{\boldsymbol{\Sigma}}^T\right) \otimes \mathbf{I}_{2L}\right]\left(\mathbf{K}_{2,2LD} \otimes \mathbf{I}_L\right)^{-1}\left\{\mathbf{I}_2 \otimes \left[\left(\mathbf{I}_{2L} \otimes \mathbf{K}_{DL}\right)\left(\text{vec } \left(\mathbf{1}_{\frac{T}{2}}^T \otimes \mathbf{I}_L\right) \otimes \mathbf{I}_D\right)\right]\right\}$$
$$+ \left[\mathbf{I}_{2L} \otimes \left(\begin{matrix} \mathbf{1}_{\frac{T}{2}}^T \otimes \mathbf{I}_L \otimes \mathbf{v}^T \\ \mathbf{1}_{\frac{T}{2}}^T \otimes \mathbf{I}_L \otimes \mathbf{w}^T \end{matrix}\right)\left(\mathbf{I}_2 \otimes \tilde{\boldsymbol{\Sigma}}\right)\right]\left(\mathbf{I}_2 \otimes \mathbf{K}_{2,L} \otimes \mathbf{I}_{LD}\right)^{-1}\left[\mathbf{I}_{4,2} \otimes \left(\mathbf{I}_L \otimes \mathbf{I}_L \otimes \mathbf{I}_D\right)\left(\text{vec } \mathbf{I}_L \otimes \mathbf{I}_D\right)\right]. \quad (27)$$

**Gradient of objective function**

Having identified the Jacobians $\mathbf{J}_q$, $\mathbf{J}_Q$, $\mathbf{J}_r$, and $\mathbf{J}_R$, the final expression for the gradient of the objective function may now be assembled:

$$
\begin{aligned}
dJ \quad = \quad & 2\left(\frac{1}{\Phi_r} - \frac{1}{\Phi_f}\right)\mathbf{v}^T\mathbf{\Sigma}(0)d\mathbf{v} \\
& -\frac{2}{\Phi_r}\mathbf{q}^T\mathbf{Q}^{-1}\mathbf{J}_q d\mathbf{v} \\
& +\frac{1}{\Phi_r}\mathrm{vec}\left(\mathbf{Q}^{-1}\mathbf{q}\mathbf{q}^T\mathbf{Q}^{-1}\right)^T\mathbf{J}_Q d\mathbf{v} \\
& +\frac{2}{\Phi_f}\mathbf{r}^T\mathbf{R}^{-1}\mathbf{J}_r\begin{pmatrix} d\mathbf{v} \\ d\mathbf{w} \end{pmatrix} \\
& -\frac{1}{\Phi_f}\mathrm{vec}\left(\mathbf{R}^{-1}\mathbf{r}\mathbf{r}^T\mathbf{R}^{-1}\right)^T\mathbf{J}_R\begin{pmatrix} d\mathbf{v} \\ d\mathbf{w} \end{pmatrix},
\end{aligned}
\tag{28}
$$

from which one identifies the gradient of the objective function as:

$$
\begin{aligned}
\nabla J = \ & 2\left(\frac{1}{\Phi_r} - \frac{1}{\Phi_f}\right)\begin{pmatrix} \mathbf{\Sigma}(0)\mathbf{v} \\ \mathbf{0} \end{pmatrix} - \frac{2}{\Phi_r}\begin{pmatrix} \mathbf{J}_q^T\mathbf{Q}^{-1}\mathbf{q} \\ \mathbf{0} \end{pmatrix} + \frac{1}{\Phi_r}\begin{pmatrix} \mathbf{J}_Q^T\mathrm{vec}\left(\mathbf{Q}^{-1}\mathbf{q}\mathbf{q}^T\mathbf{Q}^{-1}\right) \\ \mathbf{0} \end{pmatrix} \\
& + \frac{2}{\Phi_f}\mathbf{J}_r^T\mathbf{R}^{-1}\mathbf{r} - \frac{1}{\Phi_f}\mathbf{J}_R^T\mathrm{vec}\left(\mathbf{R}^{-1}\mathbf{r}\mathbf{r}^T\mathbf{R}^{-1}\right).
\end{aligned}
$$

## Forward model computation

To depict the spatial topographies of the latent components measured on the EEG and fMRI analyses, the "forward-model" [3] conveying the distribution of the latent source on the scalp (for EEG) or the cortical surface (for fMRI) was computed according to:

$$\mathbf{a}_w = \boldsymbol{\Sigma}(0)\mathbf{w} \left(\mathbf{w}^T \boldsymbol{\Sigma}(0)\mathbf{w}\right)^{-1},$$

$$\mathbf{a}_v = \boldsymbol{\Sigma}(0)\mathbf{v} \left(\mathbf{v}^T \boldsymbol{\Sigma}(0)\mathbf{v}\right)^{-1},$$

where $\mathbf{a}_w$ and $\mathbf{a}_v$ are the forward models of the driving and driven signal, respectively. The forward model expresses the level of covariance between the scalar time series $y(t)$ (or $z(t)$ in the case of the driven signal) and the observed data $\mathbf{x}(t)$.

## Regularization of block covariance matrices

Regularization of the symmetric block covariance matrix $\tilde{\boldsymbol{\Sigma}}$ was implemented by limiting its condition number to $c$, where the value of $c$ was set to $\infty$ for the simulated VAR data and $10^3$ for the EEG and fMRI datasets. Limiting the condition number was implemented by adding a small diagonal component $\sigma^2 \mathbf{I}$ to the covariance matrix, where the value of $\sigma^2 = \frac{(\lambda_1 - \lambda_{LD}c)}{c-1}$ ensures that the condition number of the covariance matrix is $c$, where $\lambda_1$ and $\lambda_{LD}$ are the largest and smallest eigenvalues of the block covariance matrix being regularized [4, 5].

## Evaluation of source estimation performance on VAR dataset

To evaluate the ability of GCA to recover the dynamics of the true latent sources, the fidelity of the recovered components was compared with those yielded by conventional approaches. To that end, the proportion of variance explained ($r^2$) was measured for the driving and driven signals of the $P = 3$ Granger pairs, as well as the first 5 PCs and first 5 ICs (ranked by signal variance). Note that the modeled system consisted of $K = 5$ latent sources, with $s_1 \rightarrow s_2 \rightarrow s_3$. Both $s_4$ and $s_5$ were statistically independent with all other latent sources. When measuring $r^2$ for GCA, $y_1$ was compared to $s_1$, $z_1$ to $s_2$, and $z_2$ to $s_3$. To assess signal fidelity for the PCs and ICs, $r^2$ was measured between $s_i, i = 1, 2, \ldots, 3$ and *all* PCs (or ICs). In each case, the value reported corresponds to the single component with the highest $r^2$ value.

The results of the comparison are shown in Fig S1, where it is clear that the signal fidelity of the GCs (right panel) significantly exceeds those yielded by PCA (left) and ICA (middle). The proportion of variance explained by the conventional methods was limited to approximately 50% and 30% for PCA and ICA, respectively. Note, however, that GCA is only able to recover sources with temporal dependencies (i.e., $s_1$, $s_2$ and $s_3$).

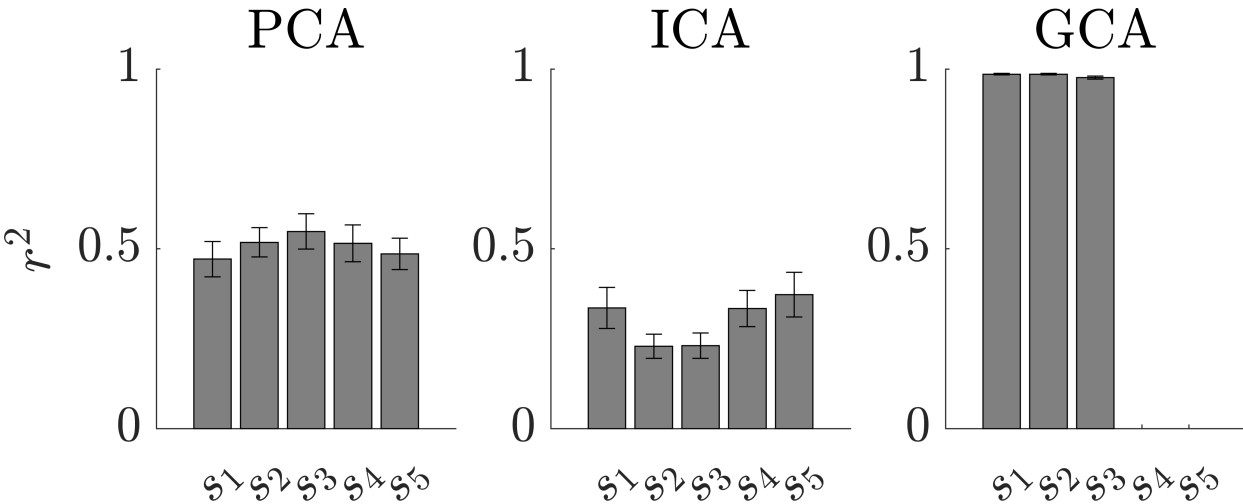

Figure S1: Comparing the source separation performance of GCA to conventional methods. In each panel, the horizontal axis depicts the identity of the latent source, while the vertical axis measures the fidelity of the recovered signal component: (left) PCA, (middle) ICA, (right) GCA. For PCA and ICA, the reported value corresponds to the component that best captures the latent signal ("arg max" across components). Error bars denote standard errors of the mean across 100 replicates.

## Power spectra of GCA components on EEG motor imagery dataset

To obtain insight into the sources recovered by the proposed method, the time series of the Granger components were transformed to the frequency domain using the multitaper spectral analysis method. The analysis employed four Slepian tapers. The spectrum was computed for all 5260 trials (spanning 52 subjects), followed by averaging of the computed spectra across trials. Panels a and b depict the spectra of 10 electrodes (5 over each hemisphere) as observed during the left and right motor imagery conditions, respectively. The electrodes were selected to cover the expected regions of activity, namely the frontal, frontocentral, central, centroparietal, and parietal areas. Panels c and d depict the spectra of the driving and driven Granger components of pairs 1 and 2 for left and right motor imagery, respectively.

Both the single electrodes and Granger components exhibit two pronounced peaks in the spectra: one near 2 Hz ("delta" band) and another over the 10 Hz alpha region. From panels c and d, it is evident that the driven signal of Granger pair 2 exhibits relatively low alpha power, consistent with desynchronization of the "mu" rhythm during motor activation [6]. On the other hand, the 10 Hz power of the driven signal in pair 1 exhibits relatively high alpha power. Given the ipsilateral and contralateral topographies of $z_1$ and $z_2$, respectively, this finding is consistent with the idea that pair 1 indicates inhibition of the uncued region, while pair 2 expresses activation of the cued structures.

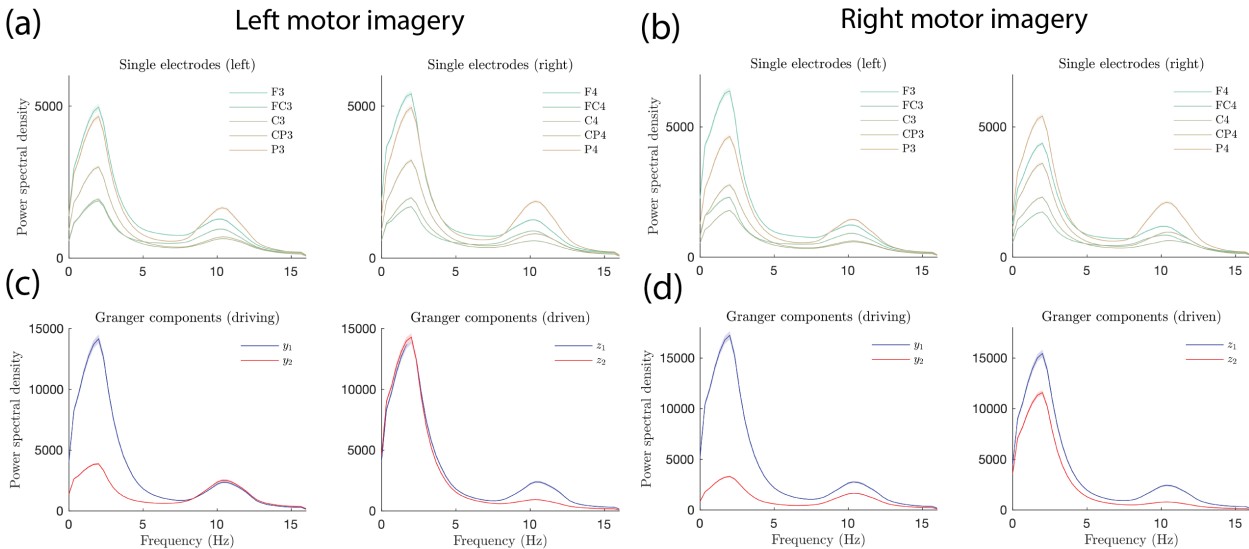

Figure S2: Power spectra of single electrodes and Granger components during motor imagery task. **(a)** Power spectra of individual electrodes over the left hemisphere (left) and right hemisphere (right), shown for the data obtained during the left motor imagery condition. **(b)** Same as (a) but now for the right motor imagery condition. **(c)** Power spectra of the driving (left) and driven (right) components of Granger pairs 1 and 2, as computed on data from the left motor imagery condition. **(d)** Same as (c) but now shown for data obtained during right motor imagery.

# Granger causality matrix on EEG left motor imagery dataset

The matrix of Granger causality strengths stemming from the right motor imagery condition is reported in the main text (see Fig 3, e-f). Fig S3 shows the corresponding result for the left motor imagery condition. The results are congruent with those from right motor imagery. Namely, the Granger causal relationships among pairs of electrodes are limited to less than $0.12$, while the relationships among the recovered Granger components are stronger (up to $0.38$ for $y_1 \rightarrow z_1$) and easier to interpret. As with right motor imagery, a relationship of the form $z_1 \leftarrow y_2 \rightarrow z_2$ is evident in the bottom row of the matrix.

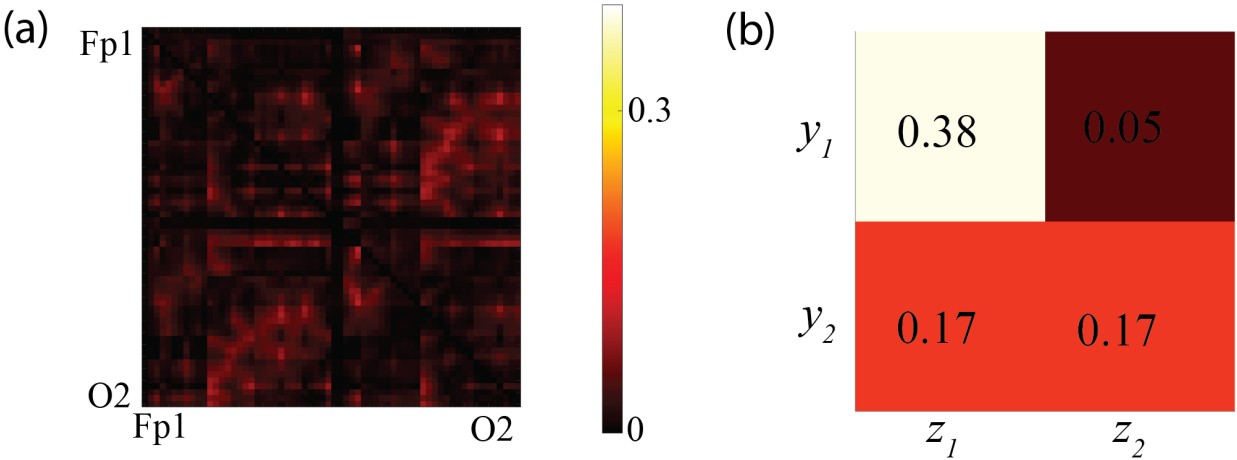

Figure S3: **(a)** Matrix of Granger causality values $(-J)$ between all pairs of electrodes on the left motor imagery condition. The values are limited to less than $0.12$ and it is difficult to identify the Granger causal structure of the system. **(b)** Same as (a) but now for the Granger components. The bottom row is suggestive of the relationship $z_1 \leftarrow y_2 \rightarrow z_2$.

## Comparison to PCA, ICA, and MVARICA on EEG motor imagery dataset

In order to determine whether the components captured by GCA on a real world dataset would also be recovered by conventional approaches, PCA, ICA, and MVARICA [7] (implemented with the code in [8]) were evaluated on the EEG motor imagery dataset described in the main text. For each technique, the first 6 components are presented.

The spatial topographies of the principal components are shown in panels a and b for the left and right motor imagery condition, respectively. The components are distinct from those recovered by GCA (compare with Fig 3 in the main text). Moreover, the PCs do not exhibit lateralization with the side of the cued hand. Note also that there is little discrimination between the PCs captured during left versus right motor imagery, which may be a consequence of the orthogonality between successive components.

Similarly, the spatial topographies of the independent components also differ from those of the Granger components (panels c and d). A clear lateralization effect is again not evident.

Finally, the components recovered by MVARICA do not resemble those of the Granger components, nor do they exhibit the expected lateralization with the side of the cued hand (panels e and f).

Overall, there was no evidence found that the signals extracted by the proposed method could also be captured by these three existing approaches.

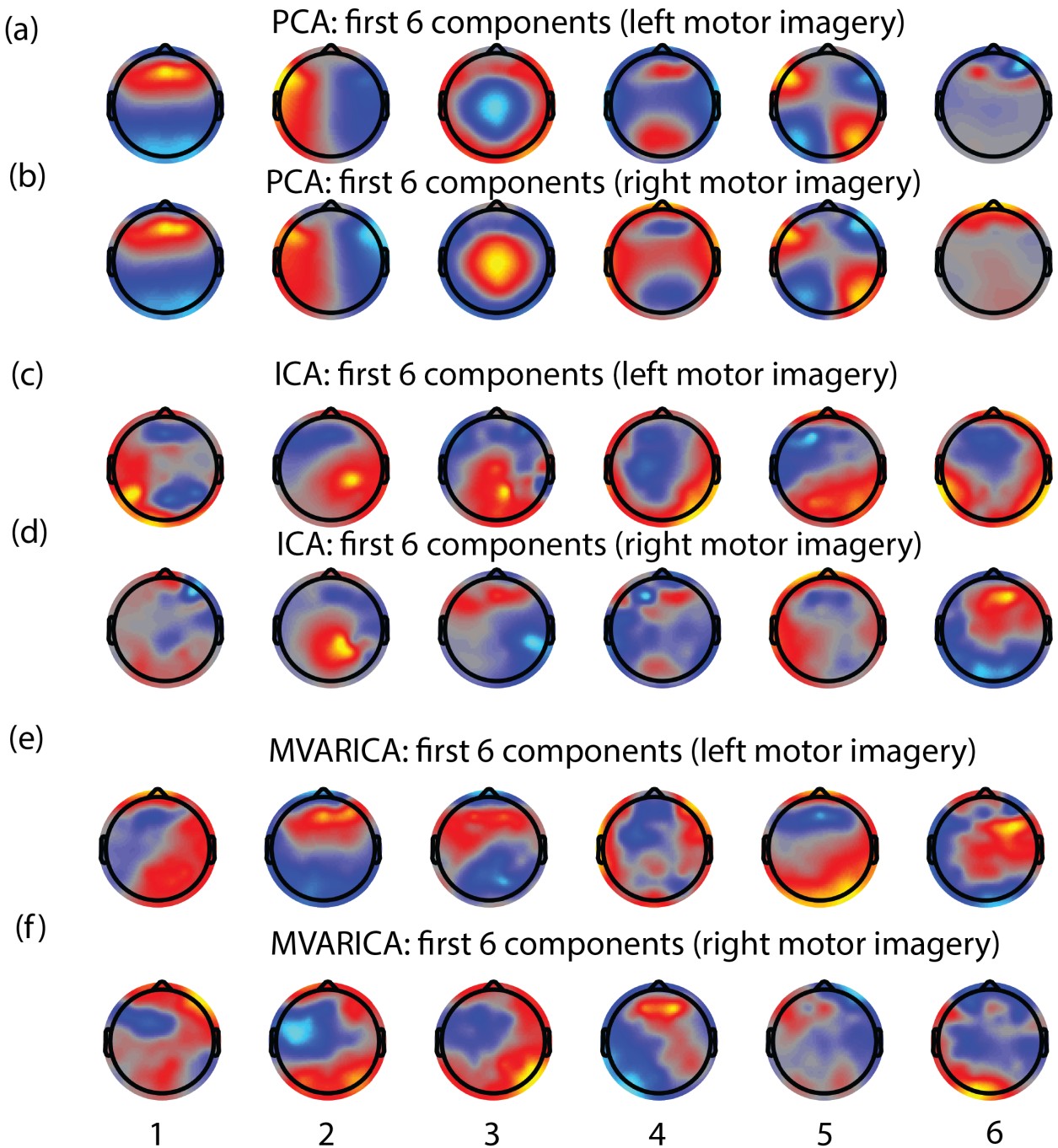

Figure S4: A comparison of the Granger components with those yielded by existing approaches. **(a)** Spatial topographies of the first six principal components, as computed on the EEG recorded during left motor imagery. **(b)** Same as (a) but now shown for right motor imagery. **(c)** The forward models of the first six independent components (ranked by signal variance), shown for the left motor imagery condition. **(d)** Same as (c) but now shown for right motor imagery. **(d)** Spatial topographies yielded by the MVARICA technique on the data from left motor imagery. **(f)** Same as (e) but now shown for the right motor imagery condition. In all rows, the spatial distributions of the depicted components are distinct from those of the driving and driven signals in the first two Granger component pairs (see Fig 3 in main text).