# OpenReview forum: "Granger Components Analysis: Unsupervised learning of latent temporal dependencies"
_NeurIPS.cc/2023/Conference — NeurIPS 2023 poster_

### Official Review · Reviewer_XpwK · 2023-07-05

**Soundness:** 3 good
**Presentation:** 2 fair
**Contribution:** 3 good
**Rating:** 5
**Confidence:** 5

**Summary:**

This paper presented a novel unsupervised learning approach using Granger Causality by identifying the driving/driven components. This method was demonstrated on EEG and fMRI data, in coincide with the neurophysiological facts.


**Strengths:**

The paper proposed an algorithm to identify the pairwise causal structure between latent variables from a multivariate observational data set and the results are supprted by simulation data and empirical data.

**Weaknesses:**

1)  The number of latent variables is a key parameter for this unsupervised learning, however it is pre-defined without any adaptive mechanism. similarly, no adaptive mechanism for L.
2) The literature is outdated and there is no performance comparison against similar algorithms for causal inference. PCA and ICA are not algorithms for causal inference.


**Questions:**

No further questions.

**Limitations:**

The simulation is limited to a specific model without variations, for example, the noise level, the model complexity, the length of the time series, etc. Makes it is difficult to assess the performance and the generaslibility of the proposed algorithm.

---

> ### Author Rebuttal · Authors · 2023-08-08
>
> ## Weaknesses
>
> ### The number of latent variables is a key parameter for this unsupervised learning, however it is pre-defined without any adaptive mechanism. similarly, no adaptive mechanism for L.
>
> We acknowledge that we did not propose an adaptive mechanism for selecting the number of latent pairs $P$, where the latent pairs are defined as $(y_i,z_i),~i=1,2,...P$. The proposed method, like PCA and CCA, produces a natural ordering of these pairs, where the first pair exhibits the largest value of Granger causality, the second pair the second largest, and so forth: $J_{y_1 \rightarrow z_1} > J_{y_2 \rightarrow z_2} > ... > J_{y_P \rightarrow z_P}$ > 0.  Similar to what is done with PCA and CCA, the number of component pairs may be selected by observing the "spectrum" of the proposed method.
>
> Please note that the objective function of the algorithm $J_{y_i \rightarrow z_i}$ is monotonically related to the "strength of causality" $G_{y_i \rightarrow z_i}$ described in the global response.
>
> For example, in the simulation results of Figure 2(b), the first two pairs yield values near 0.1, while the third is at almost zero. This indicates that $P=2$ would have been the most appropriate choice in this example. In practice, selecting a value of $P$ that is near the knee point of the data's singular value spectrum, followed by observing the $J_{y_i \rightarrow z_i}$ values returned by the proposed method, is the anticipated approach to selection of $P$.
>
> In terms of selecting $L$, this reflects a tradeoff between capturing more of the auto- and cross-correlation structure in the data and minimizing the number of model parameters, which scales with $L^2$. Knowledge of the data's autoregressive structure is helpful here: for example, for the EEG dataset, we selected $L=8$ to capture 0.5 seconds, knowing that this was about the time scale of information integration in the motor system. For the fMRI dataset, we selected $L=4$ based on the time course of the hemodynamic response function and past studies that employed this value.
>
> ### The literature is outdated and there is no performance comparison against similar algorithms for causal inference. PCA and ICA are not algorithms for causal inference.
>
> This brings about an interesting and subtle point: Granger causality is more about "prediction" than it is about true causality in the physical sense. Much of the literature on causal inference is concerned with the latter, whereas Granger causality is generally employed as a statistical assay of temporal precedence (e.g. akin to a hypothesis test). The proposed contribution of this paper, while borrowing the idea of Granger causality, is really a new approach to unsupervised component analysis. This motivated the use of classical PCA and ICA as comparison methods, which we were admittedly not 100% satisfied with. Note that methods such as Canonical Correlation Analysis require two views of a dataset, which precludes its use as a comparison here.
>
> Based on the feedback from reviewer 57Zq, we have now tested MVARICA and found that it yields surprisingly low values for the strength of causality, and components that do not flip with respect to the side of the cued hand in the EEG motor imagery dataset (please see panels e and f in the supplemental PDF).

---

> > ### Comment · Reviewer_XpwK · 2023-08-19
> >
> > Many thanks for the authors' clarification and response. It is an impressive work and I think that further comparison with the SOTA and performance evaluation is necessary since this is a very widely studied topic.

---

### Official Review · Reviewer_nBt7 · 2023-07-06

**Soundness:** 3 good
**Presentation:** 3 good
**Contribution:** 3 good
**Rating:** 6
**Confidence:** 3

**Summary:**

The paper formulates the problem of learning a pair of spatial projections that optimize a criterion based on the Granger causality between the resulting components, which is itself based on regressing the first, driving, component to the second, driven, component using a Wiener filter and the converse for the time-reversed signals. A block coordinate descent algorithm is proposed to solve for one spatial projection while the other is fixed and vice versa. Experimental results on EEG and resting state fMRI data illustrate the identification of meaningful spatial filters.



**Strengths:**

The paper is well motivated, well written, and clear.

Synthetic experiments are simple but results are convincing.

The method is applied to two different modalities of neuroimaging data and paradigms (EEG during motor imagery tasks and resting state fMRI) and the results are discussed in depth.

**Weaknesses:**

Explicit statement of the assumptions about the nature of the relationship (linear or non-linear), time-invariant, etc. is lacking. Standard Granger causality assumes a linear, time-invariant relationship. These assumptions should be stated when introducing the pairs in (2).

The algorithm could be made more succinct by removing some of the redundancies.

Minor:
Line 79 should be clarified that $\mathbf{y}_p(t)$ is also lagged form.

Line 93 variables should be defined. I assume that there exists scalar $c$ for any choice of $a$ and $b$.

On line 114, the statement 'the driven signal is not explicitly removed.' is a bit misleading. While it is not removed the explainable variance associated this component is removed by the filtering (spatiotemporal regression).

In algorithm 1, the dependency between the cost function $J$ and $\mathbf{X}$ is not explicit.

*Nit picks:* I find the capitalization of methods beyond proper nouns a bit jarring. "Granger Causlity" -> "Granger causality". "Kernel CCA" -> "kernel CCA".

Lines 70, 81 closing double quotes are wrong direction.

Lines 132–133 and 160–161 "We asked GCA" seems odd phrasing.

**Questions:**

Is automatic differentiation an option over the numerical differentiation or the manually coded derivative?

How would the method perform on synthetic data where the sources are instantaneously mixed only (like the ideal case for ICA)?

Was the use of subject versus group explored? Or was it a problem of lack of data?

Line 194: Is each ROI summarized by its mean or principal component? This is not mentioned.

**Limitations:**

A stopping criterion for estimating the number of sources is missing.

---

> ### Author Rebuttal · Authors · 2023-08-04
>
> ## Questions
>
> ### Is automatic differentiation an option over the numerical differentiation or the manually coded derivative?
>
> Automatic differentiation is an option over the manually coded derivative. By "automatic" differentiation, we mean that the gradient is computing using finite differences instead of the analytical formula. We will clarify this in the revised version.
>
> ### How would the method perform on synthetic data where the sources are instantaneously mixed only (like the ideal case for ICA)?
>
> If we understand the question correctly, this is the result of Fig 2h-k (top of page 6). We used the fully independent source case as a control to show that the method is not able to recover sources that exhibit zero Granger Causality amongst each other. We hope that this addresses the Reviewer's question.
>
> ### Was the use of subject versus group explored? Or was it a problem of lack of data?
>
> A subject-specific analysis was not explored thus far. The reason, as you guessed, was that the amount of data for each subject was relatively small relative to the dimensionality of the covariance matrices that are being estimated by the algorithm ($(LD)^2$, where $L$ = max time lag and $D$ = number of sensors). The goal of the analyses in the paper was to test the proposed method with enough data so that estimation of the covariance matrices was not obscuring the evaluation of the technique's ability to recover meaningful sources.
>
> ### Line 194: Is each ROI summarized by its mean or principal component? This is not mentioned.
>
> Thanks for catching this. It was the mean of all grey matter voxels in the ROI. We are adding this clarification to the revised manuscript.
>
> ## Weaknesses
>
> ### Explicit statement of the assumptions about the nature of the relationship (linear or non-linear), time-invariant, etc. is lacking. Standard Granger causality assumes a linear, time-invariant relationship. These assumptions should be stated when introducing the pairs in (2).
>
> Agreed -- this is being added to the revised version.
>
> ### The algorithm could be made more succinct by removing some of the redundancies.
>
> Indeed, we will revise the algorithm to make it more compact.
>
> ### Minor: Line 79 should be clarified that is also lagged form.
>
> Thanks for catching this -- we are adding the definition of ${\bf{y}}_p$ to the revised version.
>
> ### Line 93 variables should be defined. I assume that there exists scalar $c$ for any choice of $a$ and $b$.
>
> Indeed, we are adding that $a$, $b$, and $c$ are arbitrary scalars.
>
> ### On line 114, the statement 'the driven signal is not explicitly removed.' is a bit misleading. While it is not removed the explainable variance associated this component is removed by the filtering (spatiotemporal regression).
>
> Yes, good catch. The statement is incorrect (please see also the Rebuttal to Reviewer BFyu) and has been removed. Only $y_1(t)$ is being removed. This does not remove $z_1(t)$ from the data. Rather, the component of $y_1$ that is present in $z_1$ is removed with the spatiotemporal regression.
>
> ### In algorithm 1, the dependency between the cost function and is not explicit.
>
> We are adding the dependence of $J$ on the data $\bf{X}$ to the description of algorithm 1.
>
> ### Nit picks: I find the capitalization of methods beyond proper nouns a bit jarring. "Granger Causlity" -> "Granger causality". "Kernel CCA" -> "kernel CCA".
>
> We have removed the capitalization from "Causality" and "Kernel".
>
> ### Lines 70, 81 closing double quotes are wrong direction.
>
> Thanks for catching this -- it has been fixed.
>
> ### Lines 132–133 and 160–161 "We asked GCA" seems odd phrasing.
>
> Agreed. We are rephrasing this to "We estimated $P=3$ pairs of Granger components."

---

> > ### Comment · Reviewer_nBt7 · 2023-08-14
> > **after a read of the rebuttal**
> >
> > I want to thank the authors for a nice rebuttal that most of my questions.
> >
> > Automatic differentiation $\neq$ numerical differentiation. Please fix as this is misleading.

---

> > > ### Author Response · Authors · 2023-08-17
> > > **Thanks for the clarification**
> > >
> > > Thanks for catching this -- indeed, automatic differentiation is not the proper nomenclature in this case. It will be changed to "numerical".

---

### Official Review · Reviewer_57Zq · 2023-07-07

**Soundness:** 3 good
**Presentation:** 4 excellent
**Contribution:** 4 excellent
**Rating:** 7
**Confidence:** 4

**Summary:**

This paper proposes a novel (blind) source separation method that extracts pairs of components from multivariate time series between which Granger causality (GC) is maximal. This can be a very useful tool to assess direction information flow between brain areas in an unsupervised way, without having to specify the areas a-priori. The method is very elegant and implemented in a straightforward way by setting up a corresponding optimization problem and solving it via block-coordinate descent alternating between updates of the projection vectors for the sending and receiving source. Analytic gradients are provided but also autograd is reported to work well. An interesting deflation scheme is also provided whereby the sending source is projected out in each step. Thus, the same sending source cannot be found in multiple GC component pairs, but the (residual) of a sending source can play a role as either sender of receiver in a subsequently extracted pair. The method also implements time-reversal, a method to robustify GC estimates with respect to artifacts of volume conduction. A small set of simulations illustrates the convincing properties of the method, and a convincing application to motor-imagery brain-computer interface data is also provided.

**Strengths:**

The paper proposes an elegant and potentially useful method. The derivation of the method is easy to follow, apart from minor gaps. The technical parts are sound. The simulations and real data results are convincing and provide a good picture of the capabilities of the method. The writing is clear. Overall, a nice and self-contained paper.

**Weaknesses:**

The simulations and real data analyses could be better developed. The simulations could be more quantitative, e.g. studying 100 systems instead of only one, and reporting distribution of reconstruction metrics. The impact of factors such as the SNR could be systematically studied, and different types of noise could be studied. More methods could be included in the empirical comparisons. For example, BSS methods like MVARICA [1] and SCSA [2] do not assume independent components but model the sources exactly by an MVAR model, from which GC between every pair of components can be assessed. In the BCI context, the extraction of class-specific sources using CSP [3] or SMR oscillations using SSD [4] could be compared to the proposed method. Although I understand that not for all methods working code may be found.
Theoretically, it would be critical to also discuss the identifiability of the model. Linear mixtures of MVAR processes are again MVAR models and a valid question is why the maximization of GC should provide the “true” unmixing. This is especially critical as research has shown that even a mixing of independent sources can induce spurious GC [5-9]. Moreover, no non-Gaussianity of residuals as in [2] is assumed to guide the reconstruction.

[1] https://www.sciencedirect.com/science/article/abs/pii/S1053811908008549
[2] https://ieeexplore.ieee.org/abstract/document/5466024
[3] https://ieeexplore.ieee.org/abstract/document/4408441
[4] https://www.sciencedirect.com/science/article/abs/pii/S1053811914005503
[5] https://www.sciencedirect.com/science/article/abs/pii/S1053811912009469
[6] https://www.sciencedirect.com/science/article/abs/pii/S105381191401009X
[7] https://ieeexplore.ieee.org/abstract/document/7412766
[8] https://link.springer.com/article/10.1007/s10548-016-0538-7
[9] https://www.frontiersin.org/articles/10.3389/fncom.2016.00121/full


**Questions:**

-	It would be interesting if the authors could provide more information about the extracted sources in the motor imagery real data example. Could you please plot power spectra of the sources compared to some of the relevant channels over the motor areas. Similarly, it would be interesting to see the GC or TRGC connectivity spectra of the extracted component look like in comparison to the power spectrum. For this, spectral G-causality as implemented in the MVGC toolbox could be used. Finally, the corresponding forward models could be mapped into the brain using some inverse modeling techniques, which could give further insight into the physiological relevance of the extracted sources. [10] analyze functional connectivity before and during motor imagery using undirected FC metrics, and it could be interesting to see if the sources are similar.
-	Some of the derivations are not immediately clear to me. The step from Eqs. (3) and (5) to (6) is not completely clear. Perhaps a few intermediate steps could be added? When expanding the square of the residual, should there be no mixed terms (products of z and z_p)?
-	z_p and y_p should be defined. These are the temporally embedded multivariate versions of z and  y. In Eqs (3) and (4) these should still depend on (t)
-	line 93: this is not the only possibly undesirable case, one can also show that spurious GC and emerge from mixtures of independent sources. E.g. x_1 = s + n_1 and x_2 = s + n_2 for independent noises n_1 and n_2 can lead to GC between x_1 and x_2 in either direction.
-	Line 95 (“note that”): this is not true in that strict sense. What can be shown is that the flow from y -> z is reduced compared to the original temporal order, but there is no guarantee that it would vanish or even reverse (see [7])
-	However, it is true that, for a given unidirectional flow x -> y, subtracting the time reversed GC affects both the direction x -> y and y -> x (essentially adding a negative term for the direction y -> x), it makes sense to not only consider (and optimize) time reversed differences GCR_(x->y) = GC_(x->y) – GC_(x_rev -> y_rev) but also to directly work with the net GC between both directions: TRGC_(x->y) = GCR_(x->y) - GCR_(y->x).
-	Supplement “The forward model expresses the level of correlation”: technically it is pretty much the covariance.
- No reference for the origin of the fMRI dataset is given and no IRB approval for that study is mentioned.


**Limitations:**

The paper is overall solid; however, a theoretical discussion of the model identifiability would be indicated, also considering the proposed deflation scheme. The simulations and real data analyses could be extended.

---

> ### Author Rebuttal · Authors · 2023-08-05
>
> ## Questions
>
> ### It would be interesting if the authors could provide more information about the extracted sources in the motor imagery real data example. Could you please plot power spectra of the sources compared to some of the relevant channels over the motor areas. Similarly, it would be interesting to see the GC or TRGC connectivity spectra of the extracted component look like in comparison to the power spectrum. For this, spectral G-causality as implemented in the MVGC toolbox could be used. Finally, the corresponding forward models could be mapped into the brain using some inverse modeling techniques, which could give further insight into the physiological relevance of the extracted sources. [10] analyze functional connectivity before and during motor imagery using undirected FC metrics, and it could be interesting to see if the sources are similar.
>
> We have added power spectral plots of the GCs (computed with the multitaper method) alongside the spectra of the relevant single electrodes to the attached PDF. Of note, it is interesting that $y_2(t)$ (i.e., the driving signal of GC 2) has very large alpha power, where as the corresponding driven signal $z_2(t)$ has very low alpha power. This appears to be consistent with the driving signal being preparatory (alpha is synchronized) and the driven signal reflecting execution (alpha is desynchronized).
>
> We agree that connectivity spectra and inverse modeling of the components are very interesting in this application. The results of these analyses are not yet available due to time constraints of the rebuttal period.
>
> ### Some of the derivations are not immediately clear to me. The step from Eqs. (3) and (5) to (6) is not completely clear. Perhaps a few intermediate steps could be added? When expanding the square of the residual, should there be no mixed terms (products of z and z_p)?
>
> We derive Eq. (6) here.
>
> The target signal in the linear regression is $z(t)$. The predictors are $\textbf{z}_p(t)$. The Mean Squared Error is $\left< \epsilon_z^2 \right> = E [ ( z(t) - \textbf{h}^T \textbf{z}_p(t) )^2 ]$ where $\textbf{h}$ is the filter predicting $z(t)$ from $\textbf{z}_p(t)$. We would like to first identify the filter that minimizes the MSE:
>
> $\textbf{h}^{\ast} = \mathrm{arg~min}_{\textbf{h}} ~ \left< \epsilon_z^2 \right> $
>
> From the definition of the Wiener filter, this given by:
>
> $\textbf{h}^{\ast} = \textbf{Q}^{-1} \textbf{q} $
>
> where $\textbf{Q}=E [\textbf{z}_p(t) \textbf{z}_p^T(t)] $ and $\textbf{q}=E [\textbf{z}_p(t) z(t)] $.
>
> The value of the residual when employing the optimal Wiener filter is:
>
> $\epsilon_r = z(t) - {\textbf{h}^{\ast}}^T \textbf{z}_p(t) = z(t) - \textbf{q}^T \textbf{Q}^{-1} \textbf{z}_p(t) $
>
> Taking the squared expectation, we obtain the expression for the minimum mean squared error (MMSE):
>
> $\left< \epsilon_r^2 \right> = E [ \left( z(t) - \textbf{q}^T \textbf{Q}^{-1} \textbf{z}_p(t) \right) \left( z(t) - \textbf{q}^T \textbf{Q}^{-1} \textbf{z}_p(t) \right) ] $
>
> $\left< \epsilon_r^2 \right> = E [ z^2(t) - 2 \textbf{q}^T \textbf{Q}^{-1} z(t) \textbf{z}_p(t) + \textbf{q}^T \textbf{Q}^{-1}  \textbf{z}_p(t) \textbf{z}_p^T(t)  \textbf{Q}^{-1} \textbf{q}    ] $
>
> $\left< \epsilon_r^2 \right> = E [ z^2(t) ] - 2 \textbf{q}^T \textbf{Q}^{-1} E [ z(t) \textbf{z}_p(t) ] + \textbf{q}^T \textbf{Q}^{-1} E [  \textbf{z}_p(t) \textbf{z}_p^T(t) ] \textbf{Q}^{-1} \textbf{q}     $
>
> $\left< \epsilon_r^2 \right> = \sigma_z^2 - 2 \textbf{q}^T \textbf{Q}^{-1} \textbf{q}+ \textbf{q}^T \textbf{Q}^{-1} \textbf{Q} \textbf{Q}^{-1} \textbf{q}     $
>
> $\left< \epsilon_r^2 \right> = \sigma_z^2 -  \textbf{q}^T \textbf{Q}^{-1} \textbf{q} $
>
> This is being added to the derivation in the Supplementary Material.
>
> ### z_p and y_p should be defined. These are the temporally embedded multivariate versions of z and y. In Eqs (3) and (4) these should still depend on (t)
>
> Thanks for catching this. It has been fixed in the revised version.
>
> ### line 93: this is not the only possibly undesirable case, one can also show that spurious GC and emerge from mixtures of independent sources. E.g. x_1 = s + n_1 and x_2 = s + n_2 for independent noises n_1 and n_2 can lead to GC between x_1 and x_2 in either direction.
>
> Interesting: a reference to this would be helpful so that we can add a note to the manuscript. Thanks.
>
> ### Line 95 (“note that”): this is not true in that strict sense. What can be shown is that the flow from y -> z is reduced compared to the original temporal order, but there is no guarantee that it would vanish or even reverse (see [7])
>
> This is very important and greatly appreciated. A more careful reading of [7] is underway and the statement is being revised.
>
> ### However, it is true that, for a given unidirectional flow x -> y, subtracting the time reversed GC affects both the direction x -> y and y -> x (essentially adding a negative term for the direction y -> x), it makes sense to not only consider (and optimize) time reversed differences GCR_(x->y) = GC_(x->y) – GC_(x_rev -> y_rev) but also to directly work with the net GC between both directions: TRGC_(x->y) = GCR_(x->y) - GCR_(y->x).
>
> This insight into time-reversed GC is very helpful and may improve the algorithm. Modified objective functions that reflect the net GC will need to be evaluated given these comments.
>
> ### Supplement “The forward model expresses the level of correlation”: technically it is pretty much the covariance.
>
> Agreed and fixed.
>
> ### No reference for the origin of the fMRI dataset is given and no IRB approval for that study is mentioned.
>
> The omission of any reference to the fMRI data was deliberate, and motivated by the double-blind review policy of neurIPS. The dataset has been previously published (citation will be provided in revised version) and all procedures were approved by the institution's IRB.
>
> (Further response limited by character limit)

---

> > ### Comment · Reviewer_57Zq · 2023-08-16
> > **Thank you for the clarifications**
> >
> > I thank the authors for their clarifications.
> >
> > At the same time I agree with the AC that it would actually be quite nice to provide a statistical approach for assessing the statistical significance of the interaction of a given pair of extracted sources at any step of the deflation. Since these are multivariate fits with multiple parameters, overfitting can occur and has to be accounted for. That is, a null distribution consistent with independent sources but for the same degree of (over)fitting should be constructed.
> >
> > Regarding my comment that instantaneous mixing of independent sources can lead to non-zero GC, I believe that the following references contain respective examples:
> >
> > Winkler, I., Panknin, D., Bartz, D., Müller, K. R., & Haufe, S. (2016). Validity of time reversal for testing Granger causality. IEEE Transactions on Signal Processing, 64(11), 2746-2760.
> >
> > Brunner, C., Billinger, M., Seeber, M., Mullen, T. R., & Makeig, S. (2016). Volume conduction influences scalp-based connectivity estimates. Frontiers in computational neuroscience, 10, 121.
> >
> > Van de Steen, F., Faes, L., Karahan, E., Songsiri, J., Valdes-Sosa, P. A., & Marinazzo, D. (2019). Critical comments on EEG sensor space dynamical connectivity analysis. Brain topography, 32, 643-654.
> >
> > The point being that if independent sources s(t) are mixed into sensor x(t) = M * s(t) via a mixing matrix M , then the coefficient matrices of an MVAR model of x(r) are also just the coefficients of the MVAR models of the sources s(t), transformed by the same matrix M. So, even if the underlying sources are independent (diagonal MVAR coefficients), the mixed sensors will have an MVAR representation with offdiagonal terms by virtue of M, indicating Granger causality.

---

> > > ### Author Response · Authors · 2023-08-17
> > > **Response to AC comment**
> > >
> > > Thank you for the feedback and for the references. When reading the explanation of how the mixing matrix introduces spurious GCs, it appears as if this property is a strong rationale for the proposed method (i.e., demixing the signals into components prior to computing GC).
> > >
> > > I don't appear to have access to the AC's comment about statistical significance (unless I am not seeing it in the system), but it certainly seems like a good idea and not difficult to add.
> > >
> > > In particular, this could be implemented using mock surrogate data [1] that retains the power spectrum of the original signals but destroys the phase (and thus GCs among the latent signals). The algorithm could then be run over ~1000 mock records to construct a null distribution of GC values against which the true value may be compared. This would be performed after each deflation, as suggested.
> > >
> > > [1] Theiler, J., Eubank, S., Longtin, A., Galdrikian, B., & Farmer, J. D. (1992). Testing for nonlinearity in time series: the method of surrogate data. Physica D: Nonlinear Phenomena, 58(1-4), 77-94.

---

### Official Review · Reviewer_BFyu · 2023-07-09

**Soundness:** 2 fair
**Presentation:** 3 good
**Contribution:** 3 good
**Rating:** 5
**Confidence:** 4

**Summary:**

The paper proposes a factorization model to extract P pairs of latent components from a multivariate time series such that in each pair one of the time series Granger causes the other. The authors apply the approach in analysing EEG and fMRI data to show meaningful conclusion.

**Strengths:**

The paper addresses an interesting problem of finding latent variables with Granger causal structure from a multivariate time series. The paper is generally well written.

**Weaknesses:**

Although the paper proposes an intriguing approach for finding latent causal structure, in general, it might be limiting since it only considers structure with pairwise time series demonstrating causal influence on each other while in practice we would expect multiple time series potentially affecting each other. Can the authors elaborate the effectiveness of the assumed latent structure a bit more?

The simulated data does not provide complete insight of the performance of the method. It might be useful to explore more realistic situation such as more time series, more intricate (conditional) causal influences among multiple time series, and/or situations where the proposed model might fail to capture causality, e.g., one time series driving more than one time series. This might be helpful in better assessing any false positive detections and false negative misses.

It will be useful to compare the proposed method to standard conditional Granger causality on the real dataset to extract the underlying causal structure and assess if it is similar to the one inferred by the proposed approach.

**Questions:**

- how are the maps in Figure 3 and 4 generated, using w and v vectors or using A matrix?
- line 114 and line 142: if y_1 and z_1 are being explicitly removed, then how can s_2 -> s_3 be identified after s_1 -> s_2? can you please elaborate?

**Limitations:**

The authors address limitation of this approach, e.g., in the context of driving time series driving only a single driven time serie
s.

---

> ### Author Rebuttal · Authors · 2023-08-03
>
> ## Questions
>
> ### Q1. How are the maps in Figure 3 and 4 generated, using w and v vectors or using A matrix?
>
> The maps are generated using the A matrix (i.e., the forward matrix). This is motivated by the literature [1,2] which argues that the forward models, by depicting the activity that is recovered by a spatial filter, lends to better interpretability. In other words, the maps depicted in Figures 3 and 4 represent the brain activity that is expressed by each of the recovered components.
>
> [1] Haufe, S., Meinecke, F., Görgen, K., Dähne, S., Haynes, J. D., Blankertz, B., & Bießmann, F. (2014). On the interpretation of weight vectors of linear models in multivariate neuroimaging. Neuroimage, 87, 96-110.
>
> [2] Parra, L. C., Spence, C. D., Gerson, A. D., & Sajda, P. (2005). Recipes for the linear analysis of EEG. Neuroimage, 28(2), 326-341.
>
> ### Q2. Line 114 and line 142: if y_1 and z_1 are being explicitly removed, then how can s_2 -> s_3 be identified after s_1 -> s_2? can you please elaborate?
>
> Only the driving signal y_1 is explicitly removed. The driven signal z_1 is *not* regressed. Apologies for the confusion -- in the statement:
>
> "This takes the form of a spatiotemporal regression such that any signals that are correlated with y_1(t) or its lagged versions y_1(t-l), l=1,...,L are removed. Given that this includes z_1(t), the driven signal is not explicitly removed",
>
> the phrase "Given that this includes z_1(t)" is not correct and will be removed from the text. The intent of the statement was that the $s_1$ component that is present in $z_1$ will be removed by regressing out $y_1$.
>
> To understand the proposed deflation scheme, consider the case of a VAR(1) system where s1→s2 and s2→s3:
>
> $s_1(t) = a s_1(t-1) + \epsilon_1(t)$
>
> $s_2(t) = b s_1(t-1) + c s_2(t-1) + \epsilon_2(t) $
>
> $s_3(t) = d s_2(t-1) + e s_3(t-1) + \epsilon_3(t)$
>
> Assuming that $y_1$ recovers $s_1$, after iteration 1 of the algorithm, we regress out [$s_1(t)$, $s_1(t-1)$], leaving us with:
>
> $\tilde{s}_1(t) = \tilde{\epsilon}_1(t)$
>
> $\tilde{s}_2(t) = f \tilde{s}_2(t-1) + \tilde{e}_2(t)$
>
> $\tilde{s}_3(t) = g \tilde{s}_2(t-1) + h \tilde{s}_3(t-1) + \tilde{e}_3(t)$
>
> where $\tilde{s}_i$ denotes the new values of the source signals resulting from the regression.
>
> Now what is left is the s2→s3 relationship.
>
> ## Weaknesses
>
> ### Although the paper proposes an intriguing approach for finding latent causal structure, in general, it might be limiting since it only considers structure with pairwise time series demonstrating causal influence on each other while in practice we would expect multiple time series potentially affecting each other. Can the authors elaborate the effectiveness of the assumed latent structure a bit more?
>
> We acknowledge that the approach assumes a specific type of signal model. On the other hand, this type of model captures some important "use-cases", for example EEG and MEG. In both these modalities, the physics of the forward problem dictates that connected sources (i.e., dipolar current sources in the cortex) are linearly mixed in the sensors due to volume conduction. Apart from encephalography, the finding of canonical resting-state networks in the GCs of the fMRI analysis suggests that this model is also appropriate in BOLD-fMRI. Thus, the proposed method appears to be well-suited to at least the most common forms of brain imaging.
>
> It is also our hope that the central idea of finding components that maximize Granger Causality can be generalized in the future to include more flexible signal models, non-linear interactions, and alternative deflation schemes.
>
> ### The simulated data does not provide complete insight of the performance of the method. It might be useful to explore more realistic situation such as more time series, more intricate (conditional) causal influences among multiple time series, and/or situations where the proposed model might fail to capture causality, e.g., one time series driving more than one time series. This might be helpful in better assessing any false positive detections and false negative misses.
>
> Agreed. Improving the simulations to consider systems with more sources, and where the assumed signal model does not hold, will be added to a revised version of the paper.
>
> ### It will be useful to compare the proposed method to standard conditional Granger causality on the real dataset to extract the underlying causal structure and assess if it is similar to the one inferred by the proposed approach.
>
> Agreed. We have added the results of conventional Granger causality to the supplemental page (see panels a and b of the accompanying PDF), where the causality matrix is shown for both the original (electrode) data (panel a, 64 x 64) as well as the recovered components (panel b, 2 x 2).  In other words, we measured conventional Granger causality on the original data, and then again on the components found by the proposed method. The results suggest that:
> - The strength of the Granger causality is much higher for the components (0.32 for $y_1 \rightarrow z_1$ and 0.18 $y_2 \rightarrow z_2$) for the example here compared to the raw electrodes (largest value is 0.12).
> - It is very difficult to infer the structure of the system from the 64-by-64 causality matrix measured on the raw electrodes.
> - It is interesting that the second driving component ($y_2$) drives *both* $z_1$ and $z_2$ strongly, indicating a relationship of the form: $z_2 \leftarrow y_1 \rightarrow z_2$.  This actually indicates that the method *is* capable of finding connections of this form in some cases, but that this requires integration of the information across multiple pairs of components.

---

> > ### Comment · Reviewer_BFyu · 2023-08-21
> > **Thank you for the clarifications**
> >
> > I would like to thank the authors for the detailed comments, and reporting the additional analysis. I believe that adding these details will improve the quality of the paper significantly.

---

### Author Rebuttal · Authors · 2023-08-07

## Author response to all Reviewers

We are grateful for the thorough reading and helpful feedback from all of the Reviewers.

Detailed responses to each Reviewer's feedback are provided separately. This response describes the figures that have been included in the additional PDF, and highlights the most salient points of the author response.

### Comparison of proposed method to standard Granger causality on the real EEG data set
*suggested by Reviewer BFyu*

The suggestion led to an interesting finding. Namely, the algorithm has identified a structure of the form $z_2 \leftarrow y_2 \rightarrow z_1$. This is evident in the causality matrix of the recovered components (listed here for the right motor imagery data and shown in panel b of the attached PDF):

$ \left( \begin{array}{cc} G_{y_1 \rightarrow z_1} & G_{y_1 \rightarrow z_2} \newline
 G_{y_2 \rightarrow z_1} & G_{y_2 \rightarrow z_2} \end{array} \right) = \left( \begin{array}{cc} 0.32 & 0.07 \newline
 0.16 & 0.18 \end{array} \right)$,

where $G$ is the strength of causality, defined as:

$G = 1-\frac{\left< \epsilon_{f} \right>^2}{\left< \epsilon_{r} \right>^2}$,

where $\epsilon_{f}$ is the residual of the full regression model and $\epsilon_{r}$ is the residual of the reduced model. $G$ is bounded between 0 and 1, with 0 indicating zero GC.

The causality matrix indicates that component $y_2$ drives both $z_1$ and $z_2$. Interestingly, $y_2$ is a component with a topography over the left frontal electrodes, whereas $z_1$ is concentrated over the right central region and $z_2$ is focused over the left central electrodes. This is consistent with a planning/premotor circuit (i.e., $y_2$) driving both activation of the cued motor circuit (the left motor cortex $z_2$) as well as *inhibiting* the ipsilateral circuit (the right motor cortex $z_1$).

This interpretation is also consistent with the power spectra of the components (*suggested by Reviewer 57Zq* and shown in panel d, lower right, of the attached PDF):
- $z_2$ has low alpha power (desynchronization = activation)
- $z_1$ has high alpha power indicating more (synchronization = inhibition)

The finding of $y_2$ driving both $z_1$ and $z_2$ contradicts one of the stated limitations in the paper:

*A limitation of Algorithm 1 is that regressing out the driving signal after each iteration prevents one from being able to identify connections of the form $s_2 \leftarrow s_1 \rightarrow s_3$*

The caveat here is that the structure $z_2 \leftarrow y_2 \rightarrow z_1$ spans *multiple* latent pairs.

To evaluate the standard approach to Granger causality, panel (a) in the attached PDF depicts the causality matrix of the individual electrode signals. The maximum value is 0.12, and it's difficult to ascertain the system structure from inspection of the matrix of causality values.

### Comparison to MVARICA
*suggested by Reviewer 57Zq*

Motivated by the suggestion to compare the proposed method to a component analysis technique that employs the VAR model, we employed the SCOT toolbox [1] to test the MVARICA technique on the real EEG dataset.

To our surprise, the causality matrix produced by the resulting components had all values $<0.01$. This may be a consequence of working with the VAR residuals to perform the component decomposition. In any case, the spatial topographies produced by this comparison technique are shown in panels (e) and (f) of the attached PDF. Although there is a clear structure in several of the components (i.e., many components are expressed over frontocentral and central electrodes), the topographies are less smooth, and more importantly, do not appear to lateralize with the side of the cued hand.

[1] Billinger, M., Brunner, C., & Müller-Putz, G. R. (2014). SCoT: a Python toolbox for EEG source connectivity. Frontiers in neuroinformatics, 8, 22.

### Panel legend
*all panels pertain to the EEG dataset*

(a) Causality matrix measured between all pairs of electrodes. Rows (columns) depict the driving (driven) signal. The value of each element is the strength of causality. The largest value on this dataset (right motor imagery) is 0.12. Shown is the causality matrix for the right motor imagery condition.

(b) Causality matrix measured between the recovered Granger components for the right motor imagery condition.

(c) Top row: power spectra measured for selected electrodes, shown for the left motor imagery condition. Spectral analysis was performed with the multitaper method with 7 Slepian tapers. The power spectra are unnormalized to facilitate comparison of the SNR between the raw electrodes and recovered components. Characteristic "bumps" are evident over the delta (1-3 Hz) and alpha (8-13 Hz) region. Bottom row: Power spectra measured for the recovered Granger components. Note the large increase in SNR compared to the individual electrodes.

(d) Same as c but now shown for the right motor imagery condition. Note that the power spectra of the Granger components are remarkably consistent with that of the left motor imagery condition.

(e) The spatial topographies of the first six components as measured by the MVARICA method on the left motor imagery dataset.

(f) Same as e but now shown for the right motor imagery condition. Note that the topographies do not clearly lateralize with the side of the cued hand.

## Response to Ethics Review
Please note that the omitted citation to the fMRI dataset was deliberate and served to protect the identity of the authors. The dataset has been previously published and all experimental procedures were approved by the institutional review board of the home institution. This will be included after the review process.

Regarding the EEG dataset that was obtained from the [GigaDB](https://www.re3data.org/repository/r3d100010478) website (for which a citation to the paper was included), the terms of use of GigaDB indicate that the data is public domain and has a CC license.

---

### Comment · Area_Chair_PPvt · 2023-08-20
**Questions on statistical validity and state-of-the-art**

Dear Authors,

Having followed the discussion so far with great interest so far, I, together with reviewers,  would like to hear more about two topics:

*  Reduced Rank Regression (RRR), that is an asymmetric form of CCA, seems quite well suited to the purpose of the paper? Why isn't it discussed ?

* The proposed approach comes with major statistical questions that are barely addressed in the discussion with Rev. XpwK: how to make a grounded statistical decision, inference, which means in the present case, how to select the rank of the solution ? The crucial point is whether the proposed approach can reliably be used "as a black box", meaning that it should not encourage cherry-picking an arbitrary solution, but rather comes with a reliable parameter setting.

If you have time, it would be great to clarify these two points.
Sincerely,

Your AC

---

> ### Author Response · Authors · 2023-08-21
> **Response to AC**
>
> Thanks so much for the feedback and pointer to RRR.
>
> (1) RRR is certainly relevant to the paper: although formulated as a regression technique (i.e., supervised projection of X onto Y), its objective is actually very similar to that of CCA (if we now treat "X" and "Y" as the two views in CCA) with an extra multiplicative term that also tries to maximize the variance of the projection of Y.
>
> A good online discussion that discusses the relationships between PCA, CCA, and RRR is [here](https://stats.stackexchange.com/questions/206587/what-is-the-connection-between-partial-least-squares-reduced-rank-regression-a), while a relevant paper is [here](https://ieeexplore.ieee.org/document/6186732).
>
> The primary difference between RRR and the proposed method is that the letter is rooted in the maximization of Granger Causality between components, whereas RRR maximizes (instantaneous) correlation and explained variance.
>
> A discussion of RRR and its connection to the proposed technique will be added to the revision.
>
> (2) Indeed, black-box functionality is important to facilitating adoption. For selecting the number of component pairs to yield, one approach is to provide, along with the fitted coefficients and objective value, the p-value of the detected latent relationship:
>
> $p_k = \frac{ \sum_n \mathcal{1} \left( J_{kn}^{\mathrm{mock}} > J_k^{\ast} \right) }{N} $
>
> where $p_k$ is the p-value of the detected relationship of the $k$th pair, $ \mathcal{1}$ is the indicator function (evaluates to 1 if the argument is true, zero otherwise), $J_{kn}^{\mathrm{mock}}$ is the objective value of the $n$th phase-scrambled mock record in the $k$th component pair,  $J_k^{\ast}$ is objective function value produced by the $k$th pair of Granger components, and $N$ is the number of mock records (typically 500 or 1000).
>
> In terms of _a priori_ selection of the maximum number of components to construct, one can consider the approaches taken by popular implementations of ICA and CCA, which generally set the number of components to somewhere near the knee point of the singular value spectrum of the observations.
>
> Finally, although seemingly not common for unsupervised learning techniques, a holdout set followed by assessment of the GC on the unseen data is a standard approach to preventing overfitting.

---

### Decision · Program_Chairs · 2023-09-21

**Decision:**

Accept (poster)

**Comment:**

This is a borderline paper.
The reviewers appreciated the conceptual novelty of the paper, as well as the well-presented estimation procedure: being able to infer dirctly a graphical model from (low-dimensional) data is indeed attractive for domains such as EEG data analysis.x
The paper is generally well written.
Tha uthors delivered interesting insights in the rebuttal, that may well enhance the original submission.

However, comparison with relatively obvious baseline such as reudce-rank-regression is missing.
Core questions about statistical validity and model selection are not handled properly (the authors suggest a non-parametric test in the rebuttal, but the validity of such tests can be tricky). Further thoughts / analysis is needed to assess whether the parametric or non-parametric test yield reliable results. As put by reviewer 57Zq, there is no control of overfitting, which is highly problematic as the number of potential source cannot be assumed to be known in practice, at least in neuroimaging.

Experiments are simplistic (at least the simulated and fMRI parts). Moreover, the fMRI dataset is not described to avoid breaking anonymity; I think that this is bad. Methodological research should be run on publicly known datasets for the sake of reproducibility: validate new models on old data, use old models to analyse new data ---if one cares about validity.

The reviewer's consensus is a weak accept.